# White Adipose Tissue Depots Respond to Chronic Beta-3 Adrenergic Receptor Activation in a Sexually Dimorphic and Depot Divergent Manner

**DOI:** 10.3390/cells10123453

**Published:** 2021-12-08

**Authors:** Eric D. Queathem, Rebecca J. Welly, Laura M. Clart, Candace C. Rowles, Hunter Timmons, Maggie Fitzgerald, Peggy A. Eichen, Dennis B. Lubahn, Victoria J. Vieira-Potter

**Affiliations:** 1Department of Nutrition and Exercise Physiology, University of Missouri, Columbia, MO 65211, USA; edqx97@mail.missouri.edu (E.D.Q.); wellyr@missouri.edu (R.J.W.); lmckky@mail.missouri.edu (L.M.C.); ccry95@mail.missouri.edu (C.C.R.); mefmcm@mail.umsl.edu (M.F.); eichenpa@missouri.edu (P.A.E.); 2Department of Biochemistry, University of Missouri, Columbia, MO 65211, USA; hktt2c@mail.missouri.edu (H.T.); LubahnD@missouri.edu (D.B.L.)

**Keywords:** adrenergic activity, browning, insulin resistance, sex differences, UCP1, estrogen signaling, adipose tissue metabolism

## Abstract

Beta-3 adrenergic receptor activation via exercise or CL316,243 (CL) induces white adipose tissue (WAT) browning, improves glucose tolerance, and reduces visceral adiposity. Our aim was to determine if sex or adipose tissue depot differences exist in response to CL. Daily CL injections were administered to diet-induced obese male and female mice for two weeks, creating four groups: male control, male CL, female control, and female CL. These groups were compared to determine the main and interaction effects of sex (S), CL treatment (T), and WAT depot (D). Glucose tolerance, body composition, and energy intake and expenditure were assessed, along with perigonadal (PGAT) and subcutaneous (SQAT) WAT gene and protein expression. CL consistently improved glucose tolerance and body composition. Female PGAT had greater protein expression of the mitochondrial uncoupling protein 1 (UCP1), while SQAT (S, *p* < 0.001) was more responsive to CL in increasing UCP1 (S×T, *p* = 0.011) and the mitochondrial biogenesis induction protein, PPARγ coactivator 1α (PGC1α) (S×T, *p* = 0.026). Females also displayed greater mitochondrial OXPHOS (S, *p* < 0.05) and adiponectin protein content (S, *p* < 0.05). On the other hand, male SQAT was more responsive to CL in increasing protein levels of PGC1α (S×T, *p* = 0.046) and adiponectin (S, *p* < 0.05). In both depots and in both sexes, CL significantly increased estrogen receptor beta (ERβ) and glucose-related protein 75 (GRP75) protein content (T, *p* < 0.05). Thus, CL improves systemic and adipose tissue-specific metabolism in both sexes; however, sex differences exist in the WAT-specific effects of CL. Furthermore, across sexes and depots, CL affects estrogen signaling by upregulating ERβ.

## 1. Introduction

Young females are protected against metabolic dysfunction (e.g., type II diabetes, obesity, cardiovascular disease) compared to age-matched males and postmenopausal females [1,2,3,4]. The COVID-19 pandemic further demonstrates that sex and obesity play critical roles in disease, as being male, and especially male and obese, is a clear risk factor for mortality [5]. The underlying physiological and biochemical mechanisms responsible for this sexual dimorphism are unknown; however, many are thought to be exerted through estrogen (E2) signaling pathways [6,7,8]. E2 primarily signals through estrogen receptor alpha (ERα) and ER beta (ERβ), both of which are expressed in adipose tissue [9,10], as well as many other cell types in both sexes [11]. Thus, understanding the role these ERs play in both males and females is vital to the study of metabolic disease.

Adipose tissue dysfunction has been linked to numerous metabolic diseases [12,13,14], is heavily influenced by E2 [15,16,17], and is the major site of E2 synthesis in males and postmenopausal females [18,19]. These sex differences in adipose tissue physiology likely contribute to the sexual dimorphism observed in metabolic disease [1,20]. Thus, adipose tissue represents an important tissue to target when treating metabolic health in both sexes. Within mammals there are multiple depots of white adipose tissue (WAT), and in rodents, the perigonadal (PGAT) and subcutaneous (SQAT) depots are commonly studied to model visceral and subcutaneous WAT, respectively. Both of these depots are capable of storing lipid and producing and secreting inflammatory cytokines and adipokines, and are important in the regulation of glucose metabolism and systemic insulin sensitivity [13,21,22]. In addition to WAT, mammals also contain smaller mitochondrial-rich depots of brown adipose tissue (BAT), the primary site of non-shivering thermogenesis. BAT expresses high levels of its signature thermogenic protein, uncoupling protein 1 (UCP1), which we have shown repeatedly to be metabolically beneficial in both males and females [23,24]. Moreover, we and others have shown robust sex differences in the basal expression of WAT UCP1, with females expressing higher levels [9,25,26].

Our prior work has shown that the activation of the beta-3 adrenergic receptor (β3AR) in adipose tissue via the chemical ligand CL316,243 (CL) is sufficient to improve WAT and systemic metabolism, at least in females, even in the setting of a loss of ERα signaling [27]. ERα signaling mediates the feminizing [28] and potential cancer-promoting [29] effects of E2, but its absence leads to significant obesity and metabolic dysfunction [7,30]. We have previously found that CL treatment completely reverses that dysfunction [27]. CL improves glucose homeostasis by increasing the metabolic activity of adipocytes [31,32,33]. Alternatively, impaired adrenergic signaling in male adipocytes has also been implicated in the pathophysiology of obesity, diabetes, and age-related impairments in metabolism [34,35]. CL activates BAT [36], and causes “browning” within WAT [33,37], a phenomenon that involves increased mitochondrial biogenesis and the induction of UCP1. Our previous work using UCP1-null mice has highlighted the physiological importance of UCP1 in the maintenance of normal mitochondrial cellular function, and its role in systemic metabolic health [38]. This is also true under conditions of ovariectomy, where a lack of UCP1 exacerbates metabolic dysfunction [24].

While a major outcome of chronic CL treatment is an increase in the expression of UCP1 in WAT [27,33], the mechanism of this action is unclear. The effects of CL on adipocytes mirror the effects of exercise, especially regarding exercise-mediated increases in UCP1 [39,40]. Thus, it is interesting to note that there are also sex differences in exercise-mediated fat loss [41,42]. Such differences may be due to differences in ER signaling. Here, we sought to compare sexes and specific WAT depots for responsiveness to WAT browning. To that end, we aimed to characterize, in male and female mice, the molecular changes that occur across BAT and WAT depots in response to CL. Given growing evidence that E2 affects mitochondrial function [43,44], and the emerging appreciation for a strong relationship between adipocyte mitochondrial function and insulin sensitivity [45], we aimed to determine if sex or depot-specific differences in E2 signaling may help explain sex and/or depot-specific differences in response to CL.

## 2. Materials and Methods

### 2.1. Ethics Approval

All animal husbandry and experimental procedures were carried out in accordance with the AAALAC International and approved by the University of Missouri Institutional Animal Care and Use Committee.

### 2.2. Animals and Experimental Design

At 8 weeks of age, male and female C57/BL6 mice were fed a high-sucrose, high-fat diet (HFD) consisting of 46.4% kcal from fat, 36% kcal from carbohydrate, and 17.6% kcal from protein, with a density of 4.68 kcal per gram (Test Diet, St. Louis, MO, USA, #1814692) in order to induce obesity and metabolic dysfunction. Separate groups of mice were fed a normal chow (CHOW) diet consisting of 13% kcal from fat, 57% kcal from carbohydrate, and 30% kcal from protein, with an energy density of 3.3 kcal per gram (LabDiet, St. Louis, MO, USA, #5001) as a comparison to confirm HFD-induced obesity in experimental mice. Animals were pair-housed (within group) at 28 °C (i.e., thermoneutral conditions), in a light cycle from 0700 to 1900. Prior to and following drug treatments, the mice were assessed for body composition and glucose tolerance. Subsequently, the HFD-fed obese animals began a 2-week regiment of daily intraperitoneal injections of CL316,243 (#C5976, Sigma-Aldrich, dose: 1 μg/g body weight) or an equal volume of saline vehicle (CTRL). Injections occurred at 8 am each morning during the 2-week treatment period. The four HFD-fed treatment groups (i.e., Female-CTRL, Female-CL, Male-CTRL, Male-CL; *n* = 10–15/group) were euthanized at ~24 weeks of age following a 5 h fast. Blood and tissues (PGAT, SQAT, BAT and liver) were collected, weighed, and either snap-frozen in liquid nitrogen and stored at -80 °C until analysis, or, for histology, fixed in 10% formalin. Figure 1 depicts the study timeline.

### 2.3. Body Composition and Tissue Weights

Body composition, including % lean and % fat mass, was measured by a nuclear magnetic resonance imaging whole-body composition analyzer (EchoMRI 4 in 1/1100; Echo Medical Systems, Houston, TX, USA). Upon sacrifice, subcutaneous (inguinal) WAT (SQAT), visceral (perigonadal) WAT (PGAT), interscapular brown adipose tissue (BAT), and liver tissues were weighed to assess body composition and fat distribution.

### 2.4. Energy Intake and Expenditure Assessments

Indirect calorimetry was utilized before and during CL treatment (i.e., 1 week following the start of treatment). Briefly, animals were placed in indirect calorimetry chambers (Promethion; Sable Systems International, Las Vegas, NV, USA) to assess metabolic activity parameters, including total energy expenditure (TEE), resting energy expenditure (REE) and respiratory quotient (RQ). For TEE and REE, body weight was used as a covariate. Using the same metabolic cage system, spontaneous physical activity (SPA) was measured by the summation of x-, y-, and z-axis beam breaks. Each run captured at least two light and two dark cycles of each variable. Body weight and food intake were recorded on a weekly basis throughout the intervention.

### 2.5. Glucose Tolerance Testing (GTT)

Glucose tolerance tests were performed twice on animals; once after ~10 weeks of HFD and once after 1 week of CL treatment, ~1 week before sacrifice. After a 5 h fast, blood glucose was measured from the tail vein (i.e., time 0), and blood was sampled using a glucometer (Alpha Trak, Abbott Labs, Chicago, IL, USA). Glucose was administered (dose: 2 g/kg body weight) via intraperitoneal injection. Glucose measures were then taken 15, 30, 45, 60 and 120 min after the glucose bolus was administered. The glucose area under curve (AUC) from the baseline was calculated.

### 2.6. Fasting Blood Parameters

Plasma insulin, glucose, and non-esterified fatty acids (NEFA) were quantified by a commercial laboratory (Comparative Clinical Pathology Services, Columbia, MO, United States) using an Olympus AU680 automated chemistry analyzer (Beckman–Coulter, Brea, CA, USA) as per the manufacturer’s guidelines. The homeostasis model assessment of insulin resistance (HOMA-IR) was used as a surrogate measure of systemic insulin resistance ((fasting insulin (μU/L) × fasting glucose (mg/dL)/405.1) [46]). Adipose tissue insulin resistance (ADIPO-IR), another surrogate measure, was calculated as the product of fasting insulin (μU/L) and fasting NEFAs (mmol/L) [13].

### 2.7. Protein Isolation and Western Blotting

Protein was isolated and quantified from PGAT, SQAT and BAT as previously described [38]. Briefly, protein samples (10 µg/lane) were separated by 4–20% SDS-PAGE, transferred to polyvinylidene difluoride membranes using BioRad Trans-Turbo Blot system, blocked with non-fat dry milk, and incubated with appropriate primary antibodies overnight (Appendix A lists specific antibodies used). All three depots (PGAT, SQAT and BAT) were run individually, and subsequently, PGAT and SQAT were run together in order to allow for direct depot to depot comparisons. Blots were incubated in an appropriate secondary antibody conjugated to horseradish peroxidase, developed using Thermo Scientific SuperSignal West Femto Substrate, and images were captured using a ChemiDoc Imaging System (BioRad, Hercules, CA, USA). The intensities of individual protein bands were quantified using ImageLab (BioRad), expressed as a ratio to beta actin, and normalized to the Female-CTRL group, which was set at 1. AMPK activity was assessed via protein content analysis of its total and phosphorylated forms. AMPK phosphorylation on the Thr(172) residue was measured as an indication of AMPK activation [47], whereas phosphorylation on the Ser(485/491) residue was taken as indicative of AMPK inhibition [48].

### 2.8. RNA Extraction and Quantitative Real-Time RT-PCR

PGAT, SQAT and BAT samples were homogenized in a Qiazol solution using a tissue homogenizer (TissueLyser LT, Qiagen, Valencia, CA, USA). Total RNA was isolated according to the Qiagen’s RNeasy lipid tissue protocol and assayed using a Nanodrop spectrophotometer (Thermo Scientific, Wilmington, DE, USA) to assess purity and concentration. First-strand cDNA was synthesized from total RNA using the High-Capacity cDNA Reverse Transcription kit (Applied Biosystems, Carlsbad, CA, USA). Quantitative real-time PCR was performed as previously described using the ABI StepOne Plus sequence detection system (Applied Biosystems) [49,50]. Primers were purchased from IDT (Coralville, IA, USA) or Sigma Aldrich (St. Louis, MO, USA) and are listed in Appendix A. Beta actin was used as the housekeeping gene as cycle thresholds (CT) were not different across groups. mRNA expression was calculated using the 2^−ΔΔCT^ method, where ΔCT = Housekeeping gene CT—gene of interest CT and presented as fold-difference compared to the reference group. Due to samples being run as part of a larger study, males and female were not run on the same plate, and thus the direct comparison of male and female mRNA expression was not possible. However, the relative fold change and direction of change within each sex, in response to CL, were evaluated. mRNA levels were normalized to the sex-specific CTRL group, which was set at 1.

### 2.9. Adipocyte Cell Size Quantification

Formalin-fixed samples were processed through paraffin embedment, sectioned at 5 µm (interscapular BAT, visceral (PGAT) and subcutaneous (SQAT) WAT) and stained with H&E. Sections were evaluated via an Olympus BX34 photomicroscope (Olympus, Melville, NY, USA) and images were taken via an Olympus SC30 Optical Microscope Accessory CMOS color camera. Adipocyte size was calculated from three independent regions of the same 40× objective fields for SQAT, PGAT, and interscapular BAT depots (50 adipocytes/animal). Cross-sectional areas of the adipocytes were obtained from perimeter tracings using Image J software, as previously described [51]. An investigator blinded to the groups performed all procedures.

### 2.10. Statistical Analysis

A 2 × 2 analysis of variance (ANOVA) was used to evaluate the effects of sex (S = male vs. female) and treatment (T = CL vs. CTRL) and S×T interactions; 2 × 2 ANOVA was also used to determine the effects of sex and HFD (H = HFD vs. CHOW) and SxH interactions (prior to moving the HFD-fed animals through the intervention). Then, 3 × 2 ANOVA was used to evaluate the effects of S, T, and WAT depot (D = PGAT vs. SQAT), as well as all possible interactions. In order to accurately assess total and resting energy expenditure (TEE, REE, respectively), we used ANCOVA using body weight as a covariate. The outcome variables included physiological markers and protein and gene expression. The results of the ANOVA are displayed below each figure, with the *p*-values for the main and interaction effects, as appropriate. The main effects are also indicated on the graphs (e.g., “T” = treatment effect; “S” = sex effect). All statistically significant interactions were followed by post-hoc Tukey’s tests; significant between-group differences, if they occurred, are indicated by different letters, or “*” as appropriate. Because gene expression data were collected in such a way that sexes could not be compared, t-tests were used to determine treatment effects (within fat depot). All data are presented as mean ± standard error of the mean (SEM). For all statistical tests, significance was accepted at *p* < 0.05. All statistical analyses were performed with SPSS V25.0.

## 3. Results

### 3.1. Sex Differences in HFD-Induced Obesity

CHOW-fed animals were used in order to validate that HFD-induced obesity and metabolic dysfunction occurred. Appendix A describes the effects of HFD on body weight and composition, fat pad weights, and liver mass. As expected, when compared to sex-matched CHOW-fed control mice, chronic (i.e., 16 weeks) HFD feeding induced adverse changes in body composition (i.e., increased adiposity and liver mass) in both sexes, although males were more adversely affected. That is, males consistently had greater adiposity than females and gained more weight on HFD (sex (S) by HFD diet (H) interactions, *p* < 0.05 for final body weight and WAT pad weights).

### 3.2. Effects of CL on Body Composition

The 2-week CL treatment significantly reduced body weight, and did so equally in both sexes (T, *p* = 0.037) (Figure 2A). There was also a main effect of sex (S, *p* < 0.001), such that males were heavier than females. CL reduced both PGAT (T, *p* = 0.001) and BAT (T, *p* = 0.007) pad weights, but its effect on SQAT pad weight did not reach statistical significance (Figure 2B, *p* = 0.153); consistently, male fat pads were larger than female ones (all *p* < 0.001). CL did not significantly affect liver weight (Figure 3C), and had a tendency to reduce % fat mass (T, *p* = 0.074, trend) (Figure 2D) and increase % lean mass (T, *p* = 0.055, trend) (not shown). Importantly, CL treatment tended to reduce total fat and not total lean mass, thus positively affecting body composition. Consistently, males were fatter than females (S, *p* < 0.001). Lastly, CL decreased the average adipocyte size (T, *p* = 0.012) (Figure 2E) equally in both sexes and both depots. Representative histological images of PGAT, SQAT, and BAT are provided in Figure 2F. Overall, CL reduced body weight and improved body composition, independent of sex, which was mainly driven by decreases in visceral adiposity and the preservation of lean mass.

### 3.3. Effects of CL on Energy Expenditure

In neither sex did CL treatment affect total energy intake (Figure 3A), energy intake relative to body weight (Figure 3B), physical activity (Figure 3C), or respiratory quotient (i.e., fat oxidation) (Figure 3D). However, CL did increase resting (*p* = 0.001) (Figure 3E) and total energy expenditure (T, *p* = 0.004) (Figure 3F), which were adjusted for the effect of body weight (i.e., covaried for effect of body weight (BW)). Without BW adjustment, males expended more total energy than females. When BW was entered into the ANCOVA model for resting energy expenditure, its effect was significant (*p* = 0.024), as expected. After adjusting for the effect of BW, there was no main effect of sex on resting energy expenditure (*p* = 0.793), yet there was a main effect of treatment (*p* = 0.001); the S×T interaction was not statistically significant (*p* = 0.109). When BW was entered into the ANCOVA model for total energy expenditure, its effect was again significant (*p* = 0.002), as expected. Again, once adjusting for the effect of BW, there was no main effect of sex (*p* = 0.622), yet there was a main effect of treatment (*p* = 0.004); there was no significant S×T interaction (*p* = 0.144). Males consumed less energy relative to their BW, likely because they were significantly less physically active when compared to females. To summarize, weight loss was induced via CL treatment in both sexes, and this was not explained by reduced energy intake or increased physical activity; rather, it was caused by an increase in non-physical activity energy expenditure.

### 3.4. Effects of CL on Glucose Tolerance and Adipose Tissue Insulin Sensitivity

Glucose tolerance tests were performed on mice before and after at least 1 week of CL treatment. Prior to treatment, as expected, there were no differences in glucose tolerance between mice randomized to CTRL vs. CL groups (Figure 4A,B). Again as expected, male mice were less glucose-tolerant than females (S, *p* < 0.001). In both sexes, CL treatment significantly improved glucose tolerance (i.e., decreased AUC) (T, *p* < 0.001) (Figure 4C,D) and lowered fasting glucose (Figure 4E), but did not significantly affect fasting insulin (Figure 4F) or NEFAs (Figure 4G). Overall, males had greater fasting glucose and insulin levels (S, *p* < 0.01 for both), and greater Adipo-IR (Figure 4I) (i.e., surrogate measure of adipocyte insulin resistance), but there were no sex by treatment interactions. CL tended to improve HOMA-IR in both sexes (T, *p* = 0.088) (Figure 4H), but its effect on improving Adipo-IR was not significant (Figure 4I). In addition to having a greater indication of glucose intolerance, males also had lower levels of WAT GLUT4 protein (PGAT shown in Figure 4J; SQAT shown in Figure 4K), which often correlates with adipose tissue insulin resistance [52]. However, CL did not significantly affect GLUT4. Taken together, independent of sex, the 2-week CL316,243 intervention was sufficient to improve HFD-induced glucose intolerance.

### 3.5. Sex- and Depot-Specific Effects of CL on WAT Browning and Adipokine Expression

After confirming prior work demonstrating systemic metabolic protection against HFD-induced obesity and insulin resistance among females [53], we investigated the depot-specific adipose tissue mitochondrial profile between sexes, and compared sexes for the effect of CL on these mitochondrial parameters. Female PGAT had greater protein expression of all five complexes of the mitochondrial oxidative phosphorylation (OXPHOS) complex system (S, *p* < 0.05, all) (Figure 5A). Thus, the total OXPHOS content in PGAT was significantly greater among females (Figure 5B). However, no significant sex differences were detected in the protein density of mitochondrial OXPHOS in SQAT (Figure 5C,D) or BAT (Figure 5E,F). Overall, CL increased OXPHOS content across fat depots (Figure 5A–F), without any significant sex by treatment interactions. Specifically, CL significantly increased OXPHOS total protein content in PGAT (T, *p* = 0.001) (Figure 5B) and SQAT (T, *p* = 0.032) (Figure 5D), and tended to increase BAT OXPHOS protein content (i.e., C1, C2, both *p* < 0.04) (Figure 5F), although the effects were more robust in WAT. Representative Western blot images are provided as a Appendix A.

In addition to having greater total mitochondrial OXPHOS content, female PGAT also had greater protein expression of the mitochondrial uncoupling protein, UCP1 (S, *p* < 0.001) (Figure 6A), confirming previous findings [20,26]. Interestingly, this was not the case for SQAT (Figure 6B) or BAT (Figure 6C), where males and females had similar levels of UCP1. CL increased all mitochondrial markers in all depots, and furthermore, there was a significant sex by treatment interaction for UCP1, such that female PGAT was more responsive to CL (S×T, *p* = 0.011) (Figure 6A). Similarly, female PGAT was significantly more responsive to CL in terms of increasing the mitochondrial biogenesis marker, PPARγ coactivator 1α (PGC1α) (S×T, *p* = 0.026) (Figure 6D). On the other hand, regarding CL-induced increases in SQAT PGC1α, males were more responsive (S×T, *p* = 0.046) (Figure 6E). Regarding BAT, males and females were equivocal in their response to CL (Figure 6F).

Glucose regulatory peptide 75 (GRP75) is a known mitochondrial stress response protein recently shown to be associated with adipocyte browning [54]. In this study, we show for the first time that CL significantly increases its levels in all fat depots (Figure 6G–I) (T, *p* < 0.02, all). Representative Western blot images are provided as a Appendix A.

A panel of genes was assessed across depots, and those data are summarized in Table 1. To summarize the data on mitochondrial and browning genes, CL increased PGAT UCP1 mRNA in both females (T, *p* = 0.026) and males (T, *p* = 0.012), as expected, whereas the effects in SQAT and BAT did not reach statistical significance, likely due to the high variability. Overall, the mitochondrial/browning gene profile of WAT and BAT in response to CL followed a sexually dimorphic pattern, which was depot-divergent. In PGAT, females were more responsive to CL compared to males, whereas in SQAT, males were equal to or more responsive than females. Although the gene expression data in BAT were not statistically significant (likely due to high variability), a pattern of male BAT being more responsive than female BAT was observed.

Leptin and adiponectin are adipokines that are known to associate with adipose tissue and systemic metabolic health. Females had greater adiponectin protein density in PGAT (S, *p* = 0.013) (Figure 7A), whereas males had greater protein density in SQAT (S, *p* < 0.002) (Figure 7B). There were no sex differences in adiponectin in BAT, nor did CL affect BAT adiponectin (Figure 7C). CL did not significantly increase adiponectin in WAT, although there was a trend in this direction, which approached significance in SQAT (*p* = 0.068). Those trends were also observed in gene expression, where CL increased adiponectin in both PGAT and SQAT (Table 1), whereas there were no sex differences in leptin protein density in any depot (Figure 7D–F), and CL increased its protein content in PGAT (T, *p* = 0.018) (Figure 7D) and BAT (T, *p* = 0.001) (Figure 7F). Regarding gene expression changes, CL increased leptin mRNA in female SQAT (*p* = 0.029, Table 1), but no other differences were significant. Representative Western blot images are provided as a Appendix A.

Next, we investigated markers of lipolysis and de novo lipogenesis. Overall, CL tended to increase these metabolic activity markers, but rarely were these effects statistically significant. Specifically, CL’s impact on the protein content of the β3AR in WAT (Figure 8A,B) was not significant, nor was its effect on the critical lipolytic enzyme, adipose triglyceride lipase (ATGL), in PGAT (Figure 8C); however, it did significantly increase ATGL in SQAT (Figure 8D). CL did not affect BAT ATGL (Figure 8E). Hormone-sensitive lipase (HSL) was not significantly increased by CL in any depot (Figure 8F–H), but was increased at the mRNA level in WAT (*p* = 0.028) (Table 1).

Similar to the trend toward CL increasing some lipolytic proteins, it also tended to increase the marker of de novo lipogenesis, fatty acid synthase (FAS). The effect was trending for PGAT (T, *p* = 0.07) (Figure 8I), but was significant for SQAT (T, *p* = 0.003) (Figure 8J) and BAT (T, *p* = 0.019) (Figure 8K). FAS was significantly increased at the mRNA level (*p* = 0.032) (Table 1) in male WAT, but had no effect in male BAT (Table 1), or in any adipose tissue depot in females (Table 1). In terms of sex differences in protein content, females trended toward having greater PGAT B3AR (S, *p* = 0.079) (Figure 8A) and ATGL (S, *p* = 0.09) (Figure 8C), but this effect on HSL was not significant (S, *p* = 0.14) (Figure 8F). Interestingly, females had lower SQAT HSL content (S, *p* = 0.02) (Figure 8G). FAS was greater in female SQAT (S, *p* = 0.032) (Figure 8J), but this was not significant in PGAT (S, *p* = 0.052) (Figure 8I) or BAT (S, *p* = 0.35) (Figure 8K)**.** Taken together, these data indicate that both sex and CL treatment influenced adipokine expression, lipolysis and lipogenesis; however, these effects differed by depot, and coincided with the sex-divergent depot differences in browning.

### 3.6. Sex- and Depot-Specific Effects of CL on Adipose Tissue AMPK Activation

Adenosine monophosphate (AMP)-activated protein kinase (AMPK) is a cellular regulator of energy homeostasis. AMPK stimulates energy production pathways (e.g., fatty acid oxidation) and inhibits anabolic pathways. Exercise and fasting are known activators of AMPK in adipose tissue, as are the adipokines leptin and adiponectin. AMPK activation was assessed here indirectly by measuring the relative quantity of total AMPK (tAMP) that was phosphorylated on the Thr(172) residue [47]. The quantification of total AMPK that is phosphorylated on the Ser(485/491) residue was also measured as an indirect indicator of AMPK inhibitory activity [48]. CL did not significantly change total AMPK protein content (tAMPK) (Figure 9A–C) or the relative activation state of AMPK (i.e., *p*(T)AMPK/tAMPK ratio) in any depots (Figure 9D–F). CL did, however, decrease the inhibitory state of AMPK (i.e., decreased p(S)AMPK/tAMPK ratio) in PGAT (T, *p* < 0.001) (Figure 9G). This was not true for either SQAT (Figure 9H) or BAT (Figure 9I). In conclusion, a 14-day course of daily CL treatment promoted WAT AMPK signaling pathways by decreasing the inhibitory phosphoserine of AMPK (in the 5 h-fasted state).

Sex differences in total WAT AMPK content coincided with sex differences in WAT browning, such that female PGAT had greater total AMPK protein (tAMPK) (S, *p* = 0.021) (Figure 9A), whereas tAMPK was higher in male SQAT (S, *p* < 0.001) (Figure 9B). There were no sex differences in tAMPK expression in BAT (Figure 9C). Male PGAT had a higher level of p(T)AMPK/tAMPK (i.e., activated AMPK) (S, *p* = 0.005) (Figure 9D), but this was not significant for SQAT (Figure 9E). The opposite was true in BAT (S, *p* = 0.010), where females had greater p(T)AMPK/tAMPK (Figure 9F). There were no sex differences in p(S)AMPK/tAMPK (i.e., inhibited AMPK) in any adipose tissue depot (Figure 9G–I).

### 3.7. Effects of CL on WAT Depot-Specific Estrogen Receptor (ER) Expression

ERα protein content was greater in all fat depots in females (S, *p* < 0.05, all) (Figure 10A–C) compared to males, as expected; however, there were no sex differences in ERβ (Figure 10D–F). Regarding treatment effects, CL did not change ERα protein content in PGAT (Figure 10A) or SQAT (Figure 10B), but did significantly increase BAT ERα protein content (Figure 10C). CL also increased PGAT ERα in females at the level of mRNA (T, *p* = 0.013), but this was not significant in males (Table 1). In SQAT and BAT, CL did not significantly affect ERα mRNA, although there was a tendency toward an increase in both sexes.

CL treatment increased ERβ protein content in PGAT (T, *p* = 0.051, trending toward significance) (Figure 10D), and SQAT (T, *p* = 0.019) (Figure 10E), yet did not significantly affect BAT ERβ (Figure 10F).

### 3.8. Between-Depot Comparisions of CL Responsiveness

After discovering that the sexual dimorphism in WAT response to CL was depot-specific, we conducted experiments to directly compare the adipose tissue depots (D) for their responses to CL, and made these comparisons in both sexes. There were clear sex- and depot-specific patterns in the response to CL. As shown in Figure 11A, there was a S×T×D interaction (*p* = 0.045) in PGC1α protein expression, such that, whereas the response to CL was similar in both depots in females, in males, SQAT had a greater response compared to PGAT. As shown in Figure 11B, there was a D×T interaction (*p* < 0.001), such that SQAT UCP1 was more responsive to CL compared to PGAT, and this was true in both sexes. In Figure 11C, it is shown that this same pattern was observed for the mitochondrial translocator protein GRP75 (D×T, *p* = 0.013), with SQAT being more responsive than PGAT in both sexes. We next compared the WAT depots for their CL-induced estrogen receptor expression. As shown in Figure 11D, CL significantly increased the ERβ to ERα ratio in both WAT depots, and in both sexes, equally (T, *p* = 0.012). These results demonstrate that: (1) PGAT and SQAT brown equally well in females, whereas in males, PGAT is resistant to the effects of CL, compared to SQAT, and (2) ERβ is upregulated in a sex- and depot-independent manner by chronic adrenergic activation in WAT.

## 4. Discussion

Obesity and obesity-linked metabolic dysfunction (e.g., insulin resistance) are the major contributors to some of the most prevalent cardiometabolic diseases (e.g., T2DM) globally, and affect both sexes [7,26]. Mechanistically, adipose tissue dysfunction, driven by chronic energy surplus and a sedentary lifestyle, is thought to be a root cause of this metabolic dysfunction that ultimately leads to disease. Notably, for reasons that are not completely clear, young females are largely protected from white adipose tissue (WAT) dysfunction and metabolic disease when compared to age-matched males [55,56]. On the other hand, increasing WAT metabolism (e.g., via browning) is sufficient to rescue systemic metabolic dysfunction [27,37]. Browning occurs in response to chronic adrenergic activation (e.g., via exercise or the B3AR-activating compound, CL316,243 (CL)) [57], but the molecular mechanism(s) are not fully known. However, sex may influence WAT browning [20], highlighting the potential importance of sex hormone (e.g., estrogen (E2)) signaling in this process. Indeed, the fact that striking sex differences exist in adipose tissue health and susceptibility to metabolic disease, such that females are largely protected from adipose tissue dysfunction and its related diseases [58], gives us reason to explore E2 and E2 receptor (ER) signaling as a potential mechanism for improving adipocyte metabolism. We compared young obese male and female mice for adipose tissue-specific responses to a chemical ligand, CL, which we use to mimic the effects of exercise on adipose tissue. This compound, similar to exercise, stimulates adipocyte lipolysis by activating the B3AR on the surfaces of fat cells. It also induces the browning of WAT, indicated by the robust upregulation of the mitochondrial uncoupling protein 1 (UCP1) [33,57]. We compared sexes for the adipose tissue depot-specific and systemic metabolic benefits of CL, which we administered daily for two weeks.

First, we confirmed that HFD induces obesity and metabolic dysfunction in both sexes, although males were, as expected, more adversely affected. Expected sex differences were observed in body weight and adiposity, such that males were larger with greater adiposity. Males were also less glucose tolerant overall, and had greater insulin resistance (as assessed indirectly using surrogate indexes). WAT GLUT4 content was measured as an indicator of adipose tissue insulin sensitivity, and in both WAT depots assessed, females had greater levels, as predicted. CL did not affect GLUT4 protein levels. We confirmed our prior work [27] and others [31,32,33] demonstrating that CL treatment was sufficient to treat obesity and glucose intolerance and improve body composition. Importantly, these effects occurred equally in both sexes. In both sexes, CL treatment improved body composition by reducing % fat mass without reducing total lean mass, and specifically reduced visceral adiposity and mean fat cell size. Mechanistically, this was not due to reduced energy intake, as treatment did not reduce energy intake. It was also not due to increased locomotor activity, as CL did not affect cage physical activity (i.e., SPA). CL did, however, increase total and resting energy expenditure. Since males are larger than females, to appropriately compare sexes for the independent effect of CL on energy expenditure, ANCOVA using body weight as a covariate was used. These analyses proved that CL increased energy expenditure, and did so equally in both sexes, explaining the improved body composition. In terms of glucose metabolism, CL significantly improved glucose tolerance in both sexes, yet it did not significantly affect fasting insulin or NEFA levels.

Next, we compared WAT (i.e., PGAT and SQAT) and BAT depots for the upregulation of mitochondrial and browning markers. Overall, we found similarities and differences in the main effects of CL on the different adipose tissue depots by sex. In some ways, sexes responded similarly to CL, whereas there were also clear sex-divergent depot differences. Table 2 provides a summary of the main effects of CL on the three adipose tissue depots studied, and indicates where such effects were sex-divergent. Female sex and CL treatment have been reported to increase WAT mitochondrial content [20,26]. We and others have previously demonstrated sex differences in WAT UCP1 (the hallmark of non-shivering thermogenesis [59]) expression, with females having greater levels. Here, we confirm that females have greater UCP1 expression in PGAT, but the effects in SQAT or BAT were not significant. This paralleled the OXPHOS and PGC1a findings. That is, female PGAT had a greater protein content of each of the five OXPHOS complexes, but this was not true for SQAT or BAT. Nonetheless, across depots, CL increased OXPHOS protein content. The critical findings were that PGC1a (the major marker of mitochondrial biogenesis [60]) and UCP1 both responded to CL in a sexually dimorphic and depot-dependent manner that paralleled one another. That is, whereas female PGAT was overall more responsive to the effects of CL, male SQAT was equally or more responsive to CL. To validate this, we performed additional experiments wherein we assessed depot by sex by treatment interactions (Figure 11); in so doing, we confirmed this sex-divergent depot difference in browning. These data validate findings previously reported by Kim et al. [20], who also showed that female PGAT has more browning potential than male PGAT, a sex difference likely dependent upon E2 availability [20]. That study only administered CL for up to 5 days, and did not examine obese mice or systemic metabolic dysfunction, nor did they assess ER protein or mRNA content. In males, SQAT has been shown to be more responsive to cold-induced browning compared to PGAT [61], but to our knowledge, ours is the first study to directly compare the responses of PGAT, SQAT and BAT to CL, in both sexes. We demonstrate that this depot difference is only true in males, thus challenging the current dogma in the field that inguinal SQAT is the most prone to brown. Indeed, among females, PGAT is more sensitive to CL-induced browning than SQAT is.

Another novel and particularly important finding is that CL significantly induced protein expression of the mitochondrial transport/stress response protein GRP75, which we show to be upregulated across adipose tissue depots and in both sexes. GRP75 is vital for mitochondrial function in various cell types [62,63,64], and data collected in breast cancer cell lines have revealed that ERβ-associated increases in mitochondrial proteins [65] were dependent upon GRP75. We found that GRP75 expression coincided with indicators of WAT browning. As we were writing the results of this study, GRP75 was identified as a novel browning marker in perivascular adipose tissue [54]. Our findings lends support to its important role in browning.

Adiponectin is an adipokine associated with enhanced insulin sensitivity, and is also known to associate with adipocyte mitochondrial activity and browning [66]. Here, its expression paralleled the sex differences in browning response, such that it was expressed at a higher level in female PGAT, which was more responsive to CL, and it was expressed higher in male SQAT, which was also more responsive to CL (Table 2). On the other hand, changes in leptin expression were not sex-specific; CL tended to increase leptin levels in both sexes, although the effect in SQAT was not significant. It should be noted that the sex differences described herein generally only obtained when looking at differences in protein expression, as high variability in the mRNA data likely prevented statistical differences from being observed. However, there were some striking sex differences that did not reach statistical significance that should be further explored (e.g., using RNAseq) in future studies. Since protein content better reflects functional differences, we opted to focus on those differences, while we provide the mRNA data in addition for interested readers.

In addition to increasing lipolysis and beta-oxidation, CL also has been shown, somewhat counterintuitively, to promote de novo lipogenesis in both WAT and BAT [67]. Thus, in order to determine whether differences in browning were associated with differences in adipose tissue lipolytic/metabolic proteins, adipose tissue triglyceride lipase (ATGL), hormone-sensitive lipase (HSL), and fatty acid synthase (FAS) protein expression were assessed across depots. The expression of ATGL and HSL, along with the expression of the B3AR itself, tended to be higher in female PGAT, coinciding with its enhanced sensitivity to browning. Sex differences did not exist in the expression of those proteins in BAT. Males, on the other hand, had greater HSL and lower FAS in SQAT. CL did not robustly affect the protein density of these lipolytic proteins, other than increasing ATGL content in SQAT. However, in both sexes, CL increased FAS, and this was true across depots, supporting previous findings [67].

AMPK activation mitigates HFD-induced metabolic dysfunction [68] and has been shown to be required for WAT browning [69]. Its activity also often correlates with adiponectin expression, and it has been shown to be activated by estrogen in adipose tissue [17]. Here, we show that its activity followed depot-divergent patterns. The total AMPK was higher in female PGAT, but the opposite was true in SQAT, again paralleling depot-specific sex differences for browning potential and adiponectin expression. Acute β3AR stimulation is known to increase AMPK activation, as indicated by greater p(T)AMPK/tAMPK [70,71], whereas chronic adrenergic stimulation has been shown to induce the expression of AMPK [72] (i.e., increase tAMPK). Those authors suggested that chronic CL might regulate the inhibitory activity of AMPK, as indicated by the p(S)AMPK/tAMPK ratio, which our data support. That is, CL appeared to increase AMPK activity in PGAT via disinhibition (i.e., reducing inhibitory serine phosphorylation).

We previously demonstrated: (1) ERα is not required for the beneficial effects of CL—in fact, ERα-null mice appear to be more responsive to CL [27]; (2) mice lacking full ERβ activity are completely resistant to exercise-induced browning [73]; and (3) exercise induces WAT ERβ [74]. This led to the hypothesis that ERβ is involved in the browning response. Expectedly, ERα expression was higher in females, across fat depots, whereas there were no sex differences in ERβ. We replicated our previous finding that CL induces WAT ERβ [27], and extended this finding to show that this effect is consistent across sexes and depots. The mechanisms by which CL induces ERβ are not known, but our group is actively investigating this.

It has been hypothesized that E2 is at least partly responsible for the higher browning potential of female PGAT [20]. Kim et al. used a chemical-induced ovarian failure model, and showed that the greater metabolic activity in female PGAT was abolished in this setting [20]. We suggest that the mechanism may involve ERβ. ERβ is present in mitochondria of various cell types, including MCF7 (i.e., breast cancer) cells [65,75], neurons [76], endometriotic tissues [77] and cardiomyocytes [78]. E2 has been shown to increase mitochondrial ERβ content and the transcription of the mitochondrial encoded respiratory system [75]. Furthermore, the activation of ERβ with a selective ligand appears to mimic the effects of CL on WAT browning and body composition [79]; however, its role in adipocyte mitochondria is not clear. Our previous work in rodents and humans has shown a strikingly strong correlation between ERβ and UCP1, also highlighting the potential mechanistic importance of ERβ in WAT browning. Herein, we demonstrate for the first time that CL increases the ERβ/ERα protein ratio in both WAT depots and in both sexes, supporting our hypothesis that ERβ is a sex-independent steroid receptor that may play a critical mechanistic role in mediating the tissue-specific and systemic metabolic benefits of β3AR signaling (e.g., CL or exercise).

The findings of the present study should be considered in light of its limitations. (1) CL’s ability to mitigate HFD-induced dysfunction manifested after only two weeks; however, this study did not address the longer-term effects and safety of CL, and it was conducted in rodents, not humans. Although no adverse effects of CL were ob-served, it is possible that side effects may occur after longer treatment, or that its ef-fects would not be sustained long-term. Future studies should investigate the long-term effects of CL. (2) The findings are associative, not causative. The molecular mech-anisms protecting female WAT, and those responsible for the sex differences and de-pot-divergent response to CL, should be studied in future studies. Such studies may use genetically manipulated mice or biomolecular approaches to knock out and subse-quently rescue ERβ activity. (3) HFD-induced obesity was used as the model of meta-bolic dysfunction and the control diet was standard chow (i.e., not the commercially available low-fat diet control designed to match the HFD). This diet was simply used to validate that HFD-induced metabolic dysfunction occurred in both sexes, and confirm that females show protection in this regard. (4) Although both exercise and CL are known to activate adipocyte lipolysis via β3AR activation, we did not directly com-pare exercise to CL. Future studies should do so, and may support the use of CL as an adipose tissue-specific exercise mimetic.

## 5. Conclusions

The β3AR-driven browning of WAT, through exercise or chemical induction (e.g., CL), is sufficient to improve metabolic health in the setting of obesity. Here, we show that these global beneficial metabolic effects of CL are sex-independent. Moreover, we demonstrate that the WAT-specific effects of CL mechanistically diverge by sex and WAT depot. We show that female PGAT is more responsive to CL compared to male PGAT, whereas those sex differences do not exist in SQAT or BAT. Additionally, our work provides evidence supporting the role of GRP75 in browning, which appears to be independent of sex and WAT depot. Lastly, our work demonstrates that CL robustly increases WAT ERβ expression in a sex- and depot-independent manner, highlighting its potential role in the CL-mediated benefits in WAT physiology. ERβ upregulation may constitute a sex-independent mechanism by which chronic βAR activation via exercise or CL improves WAT health.

## Figures and Tables

**Figure 1 cells-10-03453-f001:**
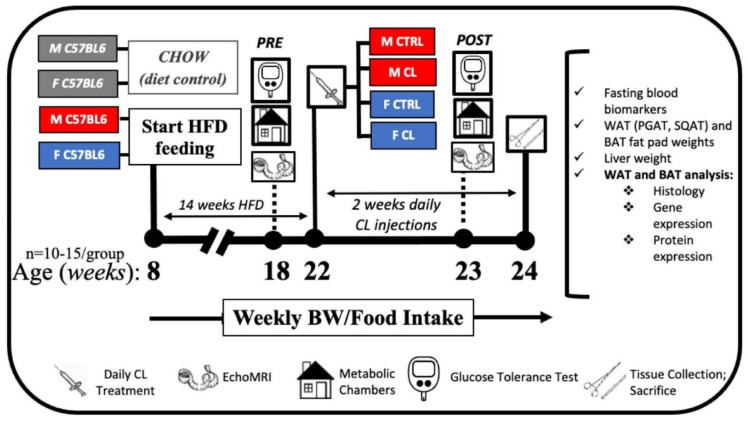
At 8 weeks of age, male (M) and female (F) C57BL/6J mice were fed standard rodent chow (CHOW) or a high-fat diet (HFD). HFD-fed mice were allowed ad libitum diet for 14 weeks in order to induce obesity before being randomized to control (CTRL) or CL treatments, creating the 4 experimental groups: M-CTRL, M-CL, F-CTRL, F-CL. The CHOW-fed male and female mice were used as a control for HFD-feeding to validate that HFD-induced obesity occurred. CHOW mice did not undergo treatments. Following obesity induction, the groups were assessed for body composition (EchoMRI), energy expenditure and physical activity (metabolic chambers), and glucose tolerance testing (GTT). Following this “PRE” testing, mice in the CL groups received daily CL316,243 injections for two weeks and mice in the CTRL groups received daily saline injections, during which time HFD feeding continued (i.e., HFD lasted for a total of 16 weeks). Prior to sacrifice, during the last week of the intervention, “POST” testing was performed prior to sacrifice. At sacrifice, blood biomarkers were measured and perigonadal (PGAT), subcutaneous (SQAT) and brown (BAT) adipose tissue depots, and whole livers, were collected and weighed. In addition, mean adipocyte cell size was measured for each fat depot. Protein expression (via western blotting) and mRNA expression (via qRT-PCR) were performed to assess depot and sex-specific responses to CL treatment.

**Figure 2 cells-10-03453-f002:**
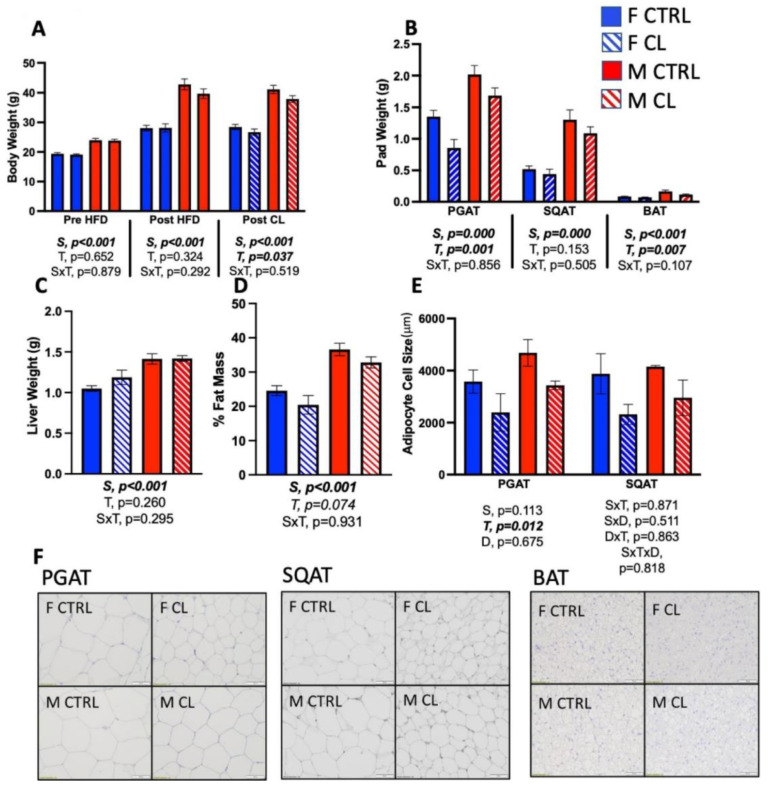
Male (M) and female (F) C57BL/6J mice, all on HFD, were given daily CL injections (1 μg/g body weight) or CTRL (saline) for 14 days, and body composition was assessed. (**A**) Average body weight at the start of the study (i.e., before HFD), post-HFD (i.e., after 14 weeks HFD feeding, before 2-week CL treatment) and post-CL (i.e., after 14 weeks HFD feeding, following 2-week CL treatment). (**B**) Pad weight for perigonadal (PGAT), subcutaneous (SQAT) and brown (BAT) adipose tissue; (**C**) liver weight; (**D**) % fat mass; (**E**) PGAT and SQAT average adipocyte size; (**F**) PGAT histological images; SQAT histological images; BAT histological images. All data are presented as mean ± SEM. *N* = 3/group for histology; *N* = 10–15/group for all other outcomes. Main effects of sex (S) and CL treatment (T) and sex by treatment interactions (S×T) were determined by 2 × 2 ANOVA and are presented below the respective figures. For adipocyte size comparisons across WAT depots (D), the main and interaction effects of depot (D), S, and T were assessed using 3 × 2 ANOVA. A *p*-value < 0.05 was accepted as significant.

**Figure 3 cells-10-03453-f003:**
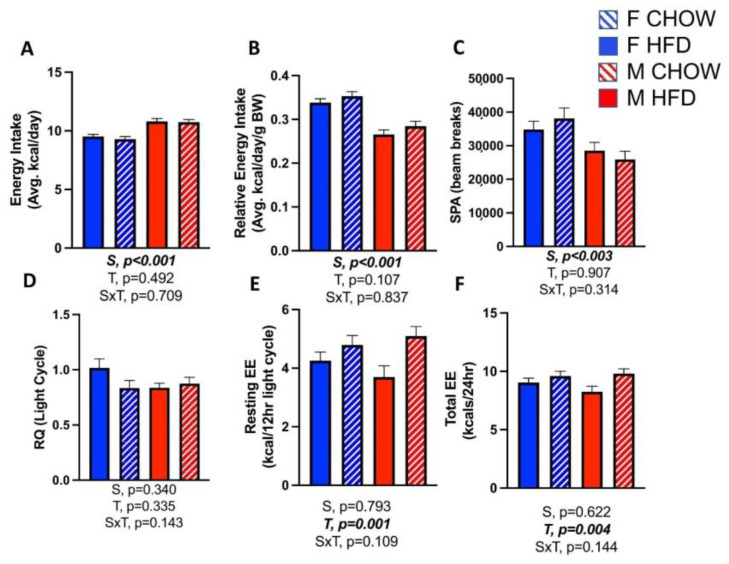
Male (M) and female (F) C57BL/6J mice, all on HFD, were treated for 2 weeks with daily CL injections (1 μg/g body weight) or CTRL (saline), and energy balance was assessed. (**A**) Average energy intake per day; (**B**) relative energy intake per day per g body weight; (**C**) spontaneous physical activity; (**D**) respiratory quotient (RQ) during light cycle; (**E**) resting energy expenditure (EE) (i.e., light/resting cycle EE covaried for body weight (BW)); (**F**) total EE covaried for BW. All data are presented as mean ± SEM. *N* = 6–10/group. Main effects of sex (S), CL treatment (T) and sex by treatment interactions (S×T) were determined by 2 × 2 ANOVA and are presented below the respective figures. *p*-value < 0.05 accepted as significant.

**Figure 4 cells-10-03453-f004:**
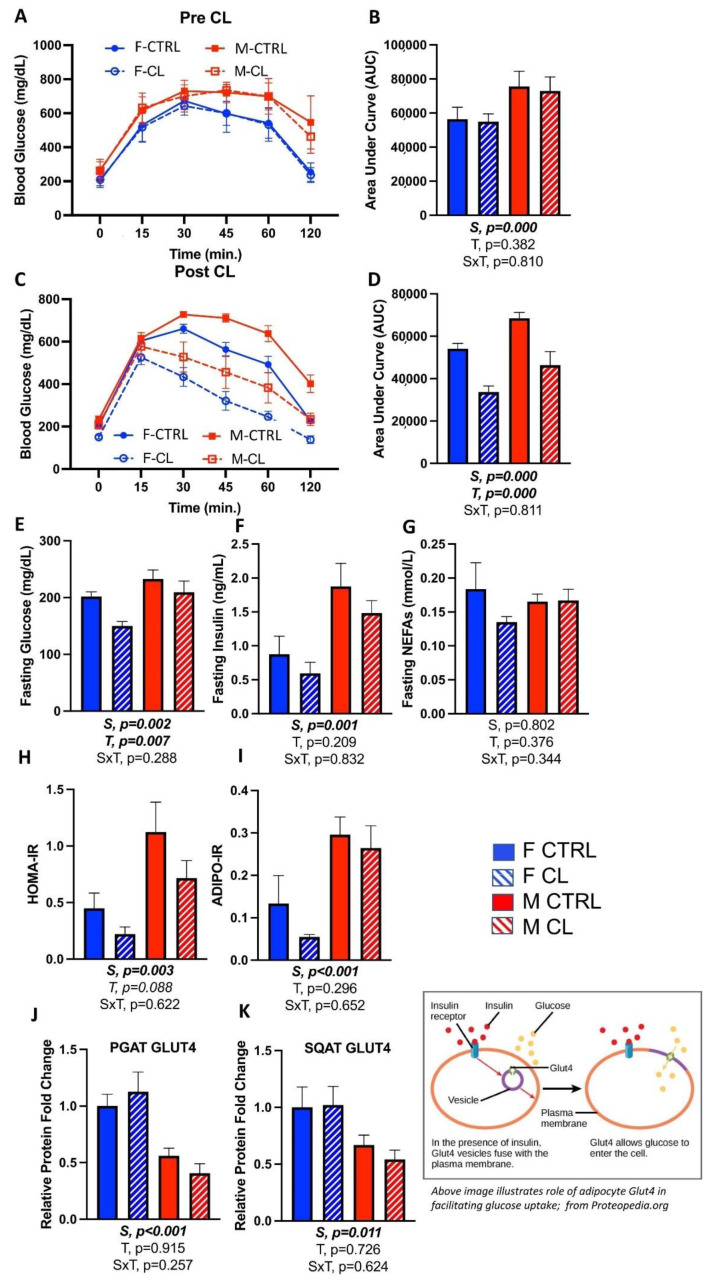
Male (M) and female (F) C57BL/6J mice, all on HFD, were treated for 2 weeks with daily CL injections (1 μg/g body weight) or CTRL (saline). Glucose tolerance tests (GTT) were performed and blood biomarkers were measured to assess the effect of CL on glucose homeostasis. (**A**) GTT graph before CL treatment; (**B**) area under curve (AUC) from GTT before CL; (**C**) GTT graph after CL treatment; (**D**) AUC from GTT after CL; (**E**) fasting blood glucose; (**F**) fasting blood insulin; (**G**) fasting blood non-esterified fatty acids (NEFAs); (**H**) Homeostatic Model Assessment for Insulin Resistance (HOMA-IR) = fasting insulin (microU/L) x fasting glucose (nmol/L)/22.5; (**I**) adipose insulin resistance (ADIPO-IR) = fasting NEFA concentration (mmol/L) x fasting insulin concentration (pmol/L); (**J**) PGAT GLUT4 relative to beta actin; (**K**) SQAT GLUT4 relative to beta actin. All data are presented as mean ± SEM. *N* = 7–14/group. Main effects of sex (S), CL treatment (T) and sex by treatment interactions (S×T) were determined by 2 × 2 ANOVA and are presented below the respective figures. *p*-value < 0.05 accepted as significant.

**Figure 5 cells-10-03453-f005:**
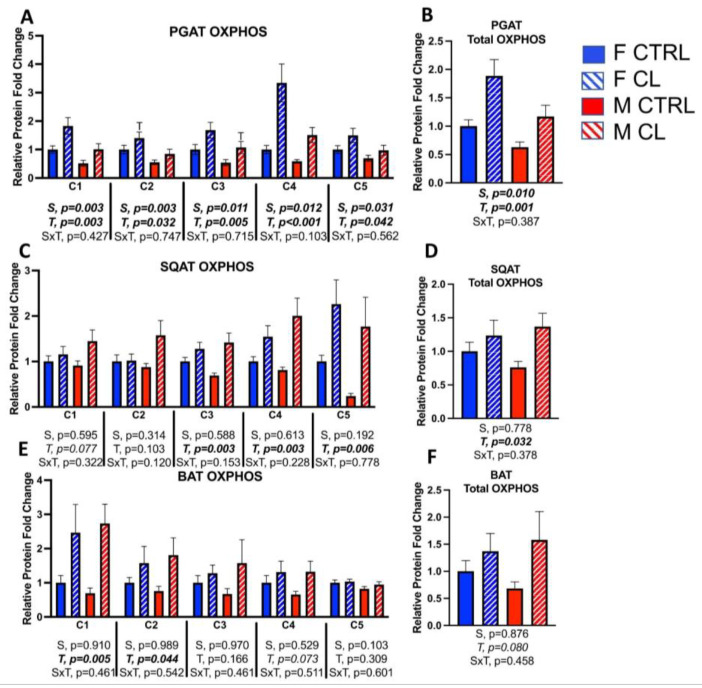
Male (M) and female (F) C57BL/6J mice, all on HFD, were treated for 2 weeks with daily CL injections (1 μg/g body weight) or CTRL (saline). Perigonadal (PGAT), subcutaneous (SQAT) and brown (BAT) adipose tissue depots were collected and the expressions of all five oxidative phosphorylation protein complexes (OXPHOS) were measured via western blot. Complex 1, NADH dehydrogenase (C1). Complex 2, Succinate dehydrogenase (C2). Complex 3, Ubiquinol Cytochrome c oxidoreductase (C3). Complex 4, Cytochrome c oxidoreductase (C4). Complex 5, ATP Synthase. Total OX PHOS represents the summation of bands C1-5. (**A**) PGAT C1-C5; (**B**) PGAT Total OXPHOS; (**C**) SQAT C1-C5; (**D**) SQAT Total OXPHOS; (**E**) BAT C1-C5; (**F**) BAT Total OXPHOS. Blot intensity relative to beta actin was calculated. All data are presented as mean ± SEM, relative to the F CTRL. *N* = 7–9/group. Main effects of sex (S), CL treatment (T) and sex by treatment interactions (S×T) were determined by 2 × 2 ANOVA, and are presented below the respective figures. *p*-value < 0.05 accepted as significant.

**Figure 6 cells-10-03453-f006:**
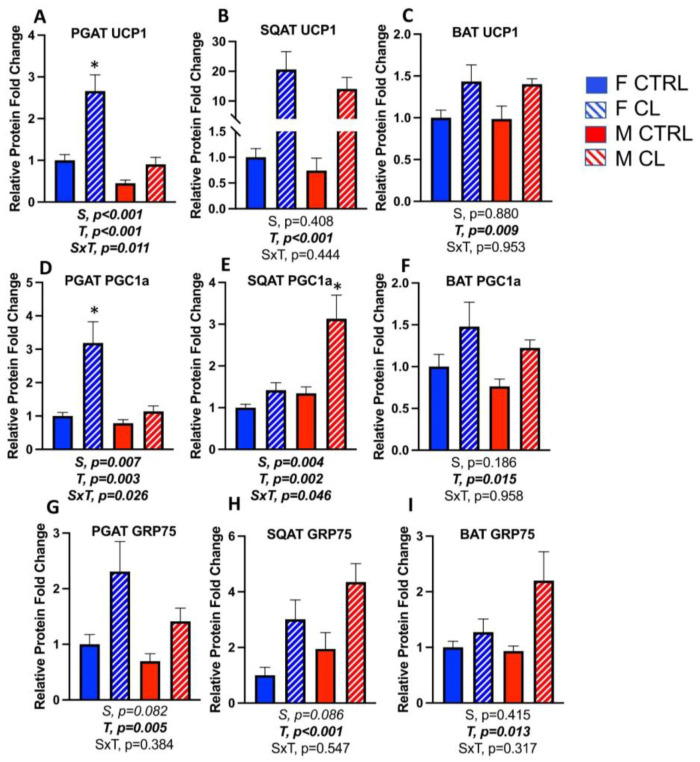
Male (M) and female (F) C57BL/6J mice, all on HFD, were treated for 2 weeks with daily CL injections (1 μg/g body weight) or CTRL (saline). Perigonadal (PGAT), subcutaneous (SQAT) and brown (BAT) adipose tissue depots were collected and the expressions of browning markers were measured via Western blot. (**A**) PGAT Uncoupling protein 1 (UCP1); (**B**) SQAT UCP1; (**C**) BAT UCP1; (**D**) PGAT PPAR-gamma coactivator 1 alpha (PGC1a); (**E**) SQAT PGC1a; (**F**) BAT PGC1a; (**G**) PGAT glucose related protein 75 (GRP75); (**H**) SQAT GRP75; (**I**) BAT GRP75. Blot intensity relative to beta actin was calculated. All data are presented as mean ± SEM, relative to the F CTRL. *N* = 7–9/group. Main effects of sex (S), CL treatment (T) and sex by treatment interactions (S×T) were determined by 2 × 2 ANOVA and are presented below the respective figures. *p*-value < 0.05 accepted as significant. * statistically different from all other groups as determined by Tukey’s post-hoc test.

**Figure 7 cells-10-03453-f007:**
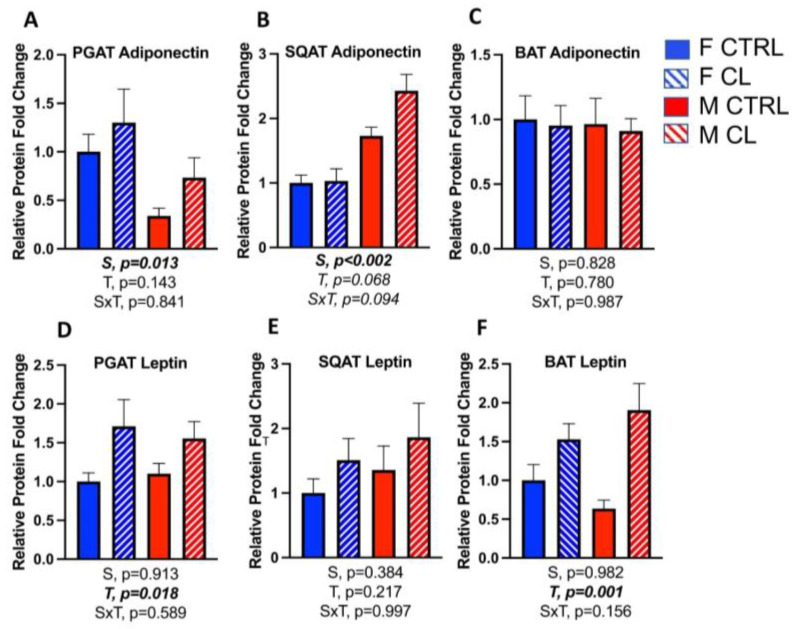
Male (M) and female (F) C57BL/6J mice, all on HFD, were treated for 2 weeks with daily CL injections (1 μg/g body weight) or CTRL (saline). Perigonadal (PGAT), subcutaneous (SQAT) and brown (BAT) adipose tissue depots were collected, and the protein of adipokines was measured via Western blotting. (**A**) PGAT adiponectin; (**B**) SQAT adiponectin; (**C**) BAT adiponectin; (**D**) PGAT leptin; (**E**) SQAT leptin; (**F**) BAT leptin. All data are presented as mean ± SEM, relative to the F CTRL. *N* = 7–9/group. Main effects of sex (S), CL treatment (T) and sex by treatment interactions (S×T) were determined by 2 × 2 ANOVA and are presented below the respective figures. *p*-values < 0.05 were accepted as significant.

**Figure 8 cells-10-03453-f008:**
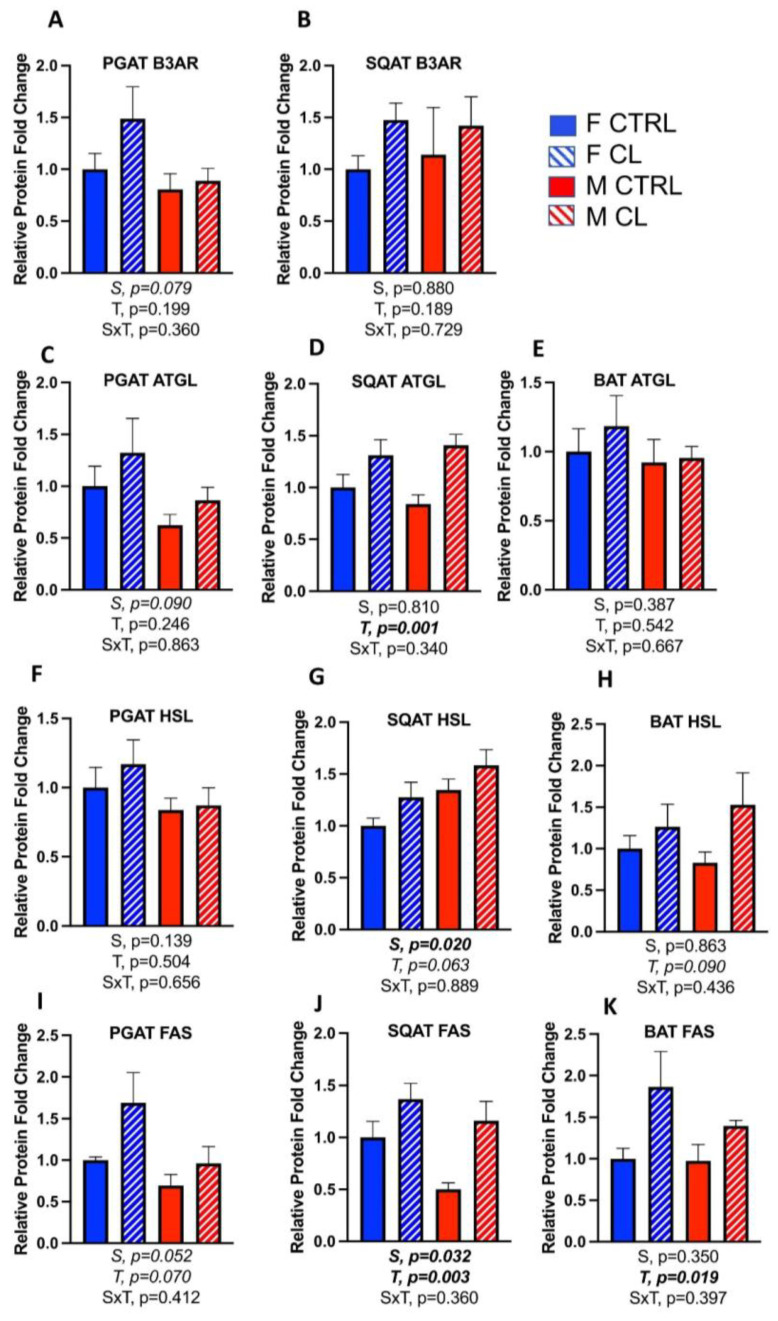
Male (M) and female (F) C57BL/6J mice, all on HFD, were treated for 2 weeks with daily CL injections (1 μg/g body weight) or CTRL (saline). Perigonadal (PGAT), subcutaneous (SQAT) and brown (BAT) adipose tissue depots were collected and markers of de novo lipogenesis and lipolysis pathways were measured via Western blot. Beta-3 Adrenergic Receptor (B3AR); Adipocyte Triglyceride Lipase (ATGL); Hormone-Sensitive Lipase (HSL); Fatty Acid Synthase (FAS). (**A**) PGAT B3AR; (**B**) SQAT B3AR; (**C**) PGAT ATGL; (**D**) SQAT ATGL; (**E**) BAT ATGL; (**F**) PGAT HSL; (**G**) SQAT HSL; (**H**) BAT HSL; (**I**) PGAT FAS; (**J**) SQAT FAS; (**K**) BAT FAS. Representative Western blot images (data expressed as relative blot intensity relative to beta actin) are provided as a Appendix A. All data are presented as mean ± SEM, relative to the F CTRL. *N* = 7–9/group. Main effects of sex (S), CL treatment (T) and sex by treatment interactions (S×T) were determined by 2 × 2 ANOVA and are presented below the respective figures. *p*-values < 0.05 were accepted as significant.

**Figure 9 cells-10-03453-f009:**
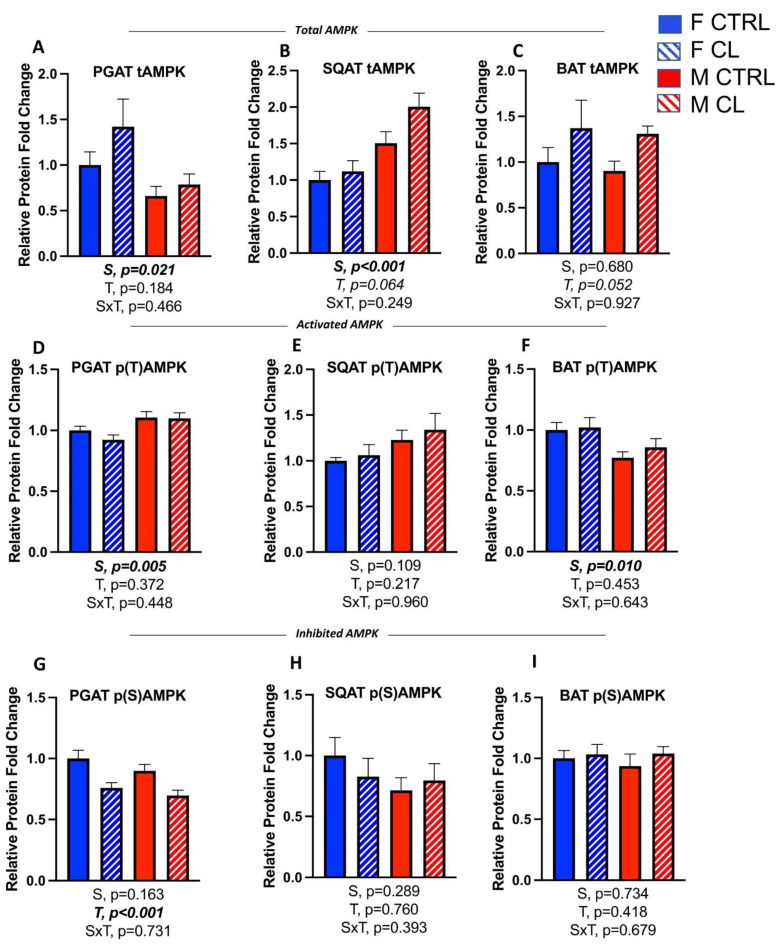
Male (M) and female (F) C57BL/6J mice, all on HFD, were treated for 2 weeks with daily CL injections (1 μg/g body weight) or CTRL (saline). Perigonadal (PGAT), subcutaneous (SQAT) and brown (BAT) adipose tissue depots were collected and the relative activation (i.e., phospho-threonine (p(S)AMPK)) and inhibition (i.e., phospho-serine (p(S)AMPK)) status of AMP Kinase (AMPK) was measured relative to total AMPK (tAMPK) via Western blot. (**A**) PGAT tAMPK; (**B**) SQAT tAMPK; (**C**) BAT tAMPK; (**D**) PGAT p(T)AMPK/tAMPK; (**E**) SQAT p(T)AMPK/tAMPK; (**F**) BAT p(T)AMPK/tAMPK; (**G**) PGAT p(S)AMPK/tAMPK; (**H**) SQAT p(S)AMPK/tAMPK; (**I**) BAT p(S)AMPK/tAMPK. Representative Western blot images are provided as a Appendix A. All data are presented as mean ± SEM, relative to the F CTRL. *N* = 7–9/group. Main effects of sex (S), CL treatment (T) and sex by treatment interactions (S×T) were determined by 2 × 2 ANOVA and are presented below the respective figures. *p*-values < 0.05 accepted as significant.

**Figure 10 cells-10-03453-f010:**
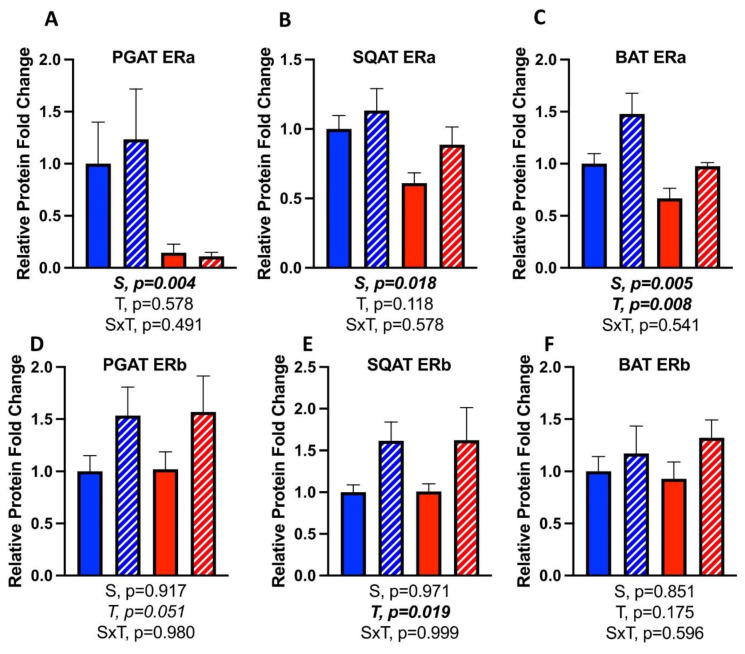
Male (M) and female (F) C57BL/6J mice, all on HFD, were treated for 2 weeks with daily CL injections (1 μg/g body weight) or CTRL (saline). Perigonadal (PGAT), subcutaneous (SQAT) and brown (BAT) adipose tissue depots were collected and the protein expression of estrogen receptor alpha (ERa) and estrogen receptor beta (ERb) was measured via Western blotting. (**A**) PGAT ERa; (**B**) SQAT ERa; (**C**) BAT ERa; (**D**) PGAT ERb; (**E**) SQAT ERb; (**F**) BAT ERb. Representative Western blot images are provided as a Appendix A. Blot intensity was expressed relative to beta actin. All data are presented as mean ± SEM. *N* = 7–9/group. Main effects of sex (S), CL treatment (T) and sex by treatment interactions (S×T) were determined by 2 × 2 ANOVA and are reported below the figure. *p*-value < 0.05 was accepted as significant.

**Figure 11 cells-10-03453-f011:**
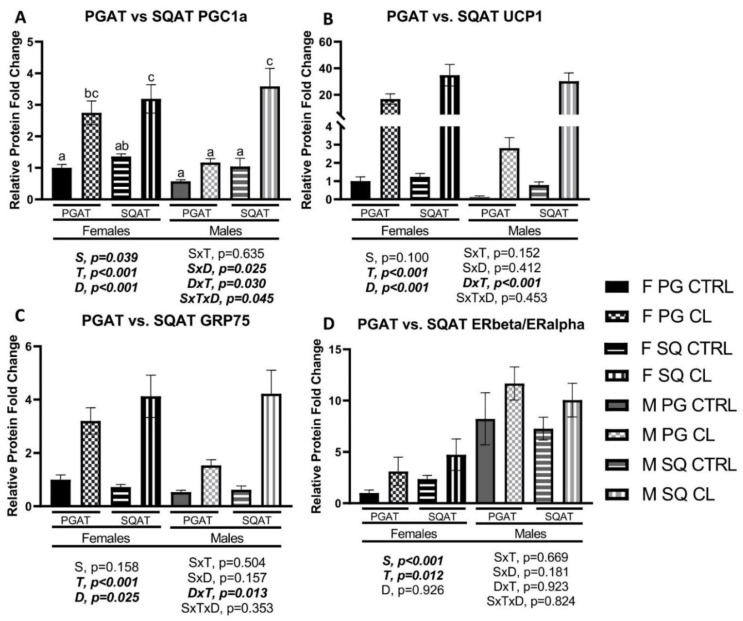
Male (M) and female (F) C57BL/6J mice, all on HFD, treated for 2 weeks with daily CL injections (1 μg/g body weight) or CTRL (saline). At sacrifice, two WAT depots (perigonadal (PGAT) and subcutaneous (SQAT)) were collected, and protein expression was assessed via Western blotting. (**A**) PPAR-gamma coactivator 1 alpha (PGC1a); (**B**) Uncoupling protein 1 (UCP1); (**C**) Glucose related protein 75 (GRP75); (**D**) Estrogen Receptor beta (ERb) to ER alpha (ERa) ratio. Representative Western blot images are provided in a Appendix A. Blot intensity was expressed relative to beta actin. All data are presented as mean ± SEM, relative to the F PGAT CTRL. *N* = 7–9/group. Main effects of sex (S), CL treatment (T), and depot (D), as well as sex by treatment (S×T), sex by depot (SxD), depot by treatment (D×T), and sex by treatment by depot (S×T×D) interactions were determined by 2 × 2 × 2 ANOVA. *p*-values < 0.05 were accepted as significant. Lower case letters distinguish statistically different groups, as determined by Tukey’s post-hoc test.

**Table 1 cells-10-03453-t001:** Male (M) and female (F) C57BL/6J mice, all on HFD, were given a 2-week CL injection (1 μg/g body wight) or CTRL (saline).

	Table 1A: Female qRT-PCR Data Table														
	PGAT CTRL		PGAT CL			SQAT CTRL		SQAT CL			BAT CTRL		BAT CL		
Gene	Mean	SEM	Mean	SEM	*p* Value	Mean	SEM	Mean	SEM	*p* Value	Mean	SEM	Mean	SEM	*p* Value
Ppargc1a	1	±0.16	7.5	±1.98	0.005	1	±0.43	24.7	±21.27	0.181	1	±0.19	1.5	±0.47	0.392
Ucp1	1	±0.62	6.4	±2.07	0.026	1	±0.55	331.1	±283.96	0.239	1	±0.15	0.9	±0.21	0.586
Hspa9	1	±0.27	3.1	±0.97	0.057	1	±0.4	207.5	±179.42	0.202	1	±0.16	0.6	±0.19	0.146
Cidea	1	±0.78	9940.0	±4346.72	0.092	1	±0.37	1.8	±0.50	0.297	1	±0.12	8.3	±7.10	0.444
Prdm16	1	±0.18	608.8	±483.46	0.283	1	±0.4	88.4	±59.81	0.117	1	±0.21	0.8	±0.23	0.563
Lipe	1	±0.33	3.5	±1.26	0.117	1	±0.21	1.0	±0.26	0.932	1	±0.17	1.0	±0.35	0.898
Fasn	1	±0.24	2.1	±0.61	0.117	1	±0.22	2.5	±1.05	0.237	1	±0.28	0.9	±0.34	0.875
Adipoq	1	±0.11	1.7	±0.19	0.067	1	±0.12	7.6	±5.83	0.342	1	±0.08	0.5	±0.17	0.042
Lep	1	±0.23	0.7	±0.21	0.329	1	±0.53	40.7	±14.78	0.029	1	±0.23	0.4	±0.13	0.058
Esr1	1	±0.13	1.7	±0.22	0.013	1	±0.14	3053.6	±1365.46	0.061	1	±0.22	1.5	±0.21	0.148
Esr2	1	±0.38	2.6	±0.79	0.091	1	±0.19	0.2	±0.09	0.005	1	±0.34	4.5	±1.40	0.044
	**Table 1B: Male qRT-PCR Data Table**														
	**PGAT CTRL**		**PGAT CL**			**SQAT CTRL**		**SQAT CL**			**BAT CTRL**		**BAT CL**		
**Gene**	**Mean**	**SEM**	**Mean**	**SEM**	***p* Value**	**Mean**	**SEM**	**Mean**	**SEM**	***p* Value**	**Mean**	**SEM**	**Mean**	**SEM**	***p* Value**
Ppargc1a	1	±0.87	0.8	±0.67	0.853	1	±0.89	3.7	±2.29	0.35	1	±0.44	2559.3	±2392.59	0.369
Ucp1	1	±0.34	52.8	±17.48	0.012	1	±0.47	7.0	±3.29	0.096	1	±0.59	4.0	±2.38	0.281
Hspa9	1	±0.22	2.0	±0.54	0.114	1	±0.3	2.3	±0.87	0.208	1	±0.47	22.7	±19.84	0.359
Cidea	1	±0.7	2.7	±0.89	0.186	1	±0.55	0.2	±0.08	0.125	1	±0.49	141.7	±79.44	0.16
Prdm16	1	±0.4	2.3	±0.75	0.191	1	±0.23	160.6	±110.49	0.179	1	±0.85	334.4	±312.72	0.336
Lipe	1	±0.25	6.2	±1.93	0.028	1	±0.2	2.6	±0.62	0.092	1	±0.53	3.8	±1.56	0.148
Fasn	1	±0.17	2.8	±0.81	0.051	1	±0.19	2.5	±0.46	0.032	1	±0.66	7.1	±5.06	0.288
Adipoq	1	±0.21	2.4	±0.75	0.109	1	±0.15	2.2	±0.39	0.056	1	±0.58	3477.8	±3249.14	0.408
Lep	1	±0.22	1.0	±0.23	0.989	1	±0.41	2.1	±0.45	0.131	1	±0.84	337.4	±315.61	0.336
Esr1	1	±0.16	1.6	±0.36	0.159	1	±0.25	1045.7	±663.33	0.207	1	±0.35	1.9	±0.26	0.092
Esr2	1	±0.2	1.8	±0.41	0.111	1	±0.33	13.77	±12.05	0.432	1	±0.43	7.4	±3.39	0.091

Perigonadal (PGAT), subcutaneous (SQAT) and brown (BAT) adipose tissue depots were collected and the expression of genes were measured via qRT-PCR. PPAR-gamma coactivator 1 alpha (Ppargc1a); Uncoupling protein 1 (Ucp1); heat shock protein a9 (Hspa9) (i.e., GRP75); Cell Death-Inducing DFFA-Like Effector A (Cidea); PR domain containing 16 (Prdm16); hormone sensitive lipase (Lipe); Fatty acid synthase (Fasn); Adiponectin (Adipoq); Leptin (Lep); Estrogen receptor alpha (Esr1); Estrogen receptor beta (Esr2). (**A**) Female gene expression; (**B**) male gene expression. Relative mRNA expression was expressed as 2 to the power of (gene of interest – beta actin), then normalized to same-sex CTRL. All data are presented as mean ± SEM. *N* = 5–9/group. Statistically significant differences were determined via *t*-test. *p*-value < 0.05 accepted as significant.

**Table 2 cells-10-03453-t002:** Overall summary of the depot-specific effects of CL by sex *.

	PGAT	SQAT	BAT
Overall effect of CL on fat pad weight and mean adipocyte size	** 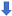 **⇓	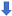 ⇔	** 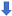 **⇓
** *Adipose tissue-specific effects of CL:* **
Mitochondrial OX PHOS content	** 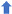 ** **⇑**	⇑	⇑
Mitochondrial biogenesis	** 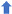 ** ⇑ ** ⇑ **	** 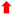 ** ** ⇑ ** ⇑	⇑
Mitochondrial uncoupling	** 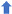 ** ⇑ ** ⇑ **	⇑	⇑
Mitochondrial GRP75 induction	**  **⇑	** 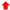 **⇑	⇑
AMPK total protein content	** 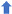 **	** 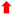 **	⇑
Net AMPK activity	⇑	⇔	
Adiponectin expression	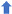	** 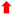 **	⇔
Leptin expression	⇑	⇔	⇑
GLUT4 expression	** 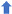 **⇔	** 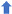 **⇔	
Lipogenesis protein (FAS) induction	**  **⇑	** 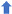 **⇑	⇑
Lipolysis protein (ATGL) induction	⇔	⇑	⇔
ERβ protein induction	⇑	⇑	⇔
ERα protein induction	** 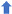 **⇔	** 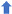 **⇔	** 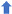 **⇑

Black arrows indicate sex-independent effects; colored arrows (female, male) represent sex-specific effects (i.e., sex interactions); bold arrows represent greater magnitude of effect; smaller arrows represent modest/trending effect; solid arrows represent sex differences at baseline. * Regarding systemic metabolism, in both males and females, CL improved body composition by reducing fat mass, preserving lean mass, and reducing visceral adiposity and mean adipocyte size. In both sexes, those improvements were attributed to increased resting energy expenditure despite no changes in energy intake or physical activity. CL also improved glucose tolerance in both sexes.

## Data Availability

All data are available upon request.

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
