# Peer review of "White Adipose Tissue Depots Respond to Chronic Beta-3 Adrenergic Receptor Activation in a Sexually Dimorphic and Depot Divergent Manner"

_cells, 2021, doi:10.3390/cells10123453_

Round 1

Reviewer 1 Report

The authors' comparison of male and female ER differences looks interesting but this paper requires revision.

Figure 3A: If you change the color or change male or female to empty, it will look better.

Figure 3: H, I, J Image quality should be raised. It's not clear.

If you organize it well, we can see qPCR data and western blot data immediately.

Pictures likely to be merged (Fig 9, 10), (Fig 11, 12), (Fig 14, 15).

Author Response

Dear reviewer 1,

Thank you for your excellent suggestions – we have improved the quality of the figures, majorly condensed figures, and consolidated PCR data to a table rather than figures. We hope this improves the focus and clarity of the work. Thank you!

Figure 3A: If you change the color or change male or female to empty, it will look better.

Thank you – we decided to use color figures instead and this greatly improved the clarity.

Figure 3: H, I, J Image quality should be raised. It's not clear.

If you organize it well, we can see qPCR data and western blot data immediately.

Pictures likely to be merged (Fig 9, 10), (Fig 11, 12), (Fig 14, 15).

Reviewer 2 Report

The authors examined the effect of CL in obesity-induced mice in a sex-specific and tissue-specific manner. The authors examined the animals physiologically (metabolism) on the one hand and the adipose tissues on both protein and RNA levels on the other hand.

The present manuscript contains a large number of results (16. figures, at least from a-e), which on the one hand, is very comprehensive but also has a detrimental effect on the clarity of the manuscript.

To improve the clarity of the manuscript, I would propose the following:

  1. One of the advantages of publishing with Cells is that you can design the layout of the text and illustrations yourself. Therefore, it would be more than advantageous if the illustrations could be found near the description of the results. Formatting of the paper or a structural adjustment would therefore be desirable.
  2. Unfortunately, the pictures are not of the best quality. In the meanwhile, I got a proper version, but the lines of the diagrams are sometimes difficult to assign (e.g. Fig. 2a, 2c, 3a, 4a, 5a, etc.). I would prefer a colored representation. I would further suggest enlarging the figures to the whole page width.
  3. For my feeling, the manuscript contains too many illustrations. The included Western blots are not discussed at any time! In my opinion, they could be moved to the supplementary part.

I would summarize the PCR data all together in one table because they are rarely significant. However, the figures shown suggest significant differences due to the use of standard errors and the partly unfavorable choice of the Y-axis interruption, which do not exist due to the high deviations between the samples. Nevertheless, I am convinced that there is a significant overall treatment effect, especially for the browning markers. This could all be noted in the table to achieve clarity.

  1. Concerning “significances”, I have problems with using letters to mark them here in this manuscript because it suggests also a comparison between i.e. F CL and M CTRL (see Fig. 7a or 7d), which makes no sense. I think it should be more clear to use lines above the bars.
  2. At two points, I had difficulty following because the methods section did not introduce me to the respective sections. That was one, the part of gender effect of HFD and CHOW on body weight, etc. Since the methods section only talks about final four groups (line 112) and this comparison is not included in Figure 1, this briefly caused confusion. I would suggest either including the evidence for the effect of HFD in the supplementary part or making additions in the M&M part and/or Figure 1. The second difficult part was the part about the activation or inhibition of AMPK. There also needs to be an explanation in the methods section.
  3. The Material and methods part should be improved at different places: a) Western Immunoblotting to give more information considering activation and inhibition of AMPK, b) animal treatment to clarify the groups, and c)The method concerning adipocyte cell size is missing.
  4. Furthermore, figure 1 is incomplete and partly contradicts the M&M part (see comment below). This figure would need to be revised.
  5. I found inhomogeneous use of abbreviations for chronic beta3 adrenergic receptor (there are three different kinds of abbreviation) and EchoMRI. Partly too many abbreviations were used anyway. Some like DNL can be saved if the abbreviation was used only once again. The abstract in particular contains 16 abbreviations, some of which were not even introduced. I would suggest to introduce the abbreviations for some of them later.
  6. Some references in the Introduction are missing.

Furthermore, I have noted mistakes/details that I have recognized (in chronological order) while reading that would also need to be improved (there may be duplications):

Line 2: Replace B3 by Beta-3 like used in the abstract

Line 19-20: please introduce the abbreviation UCP1, PGC1a, and OxPhos

Line 62-64: reference is missing

Line 65: literature is missing

Line 67: literature is missing

Line 74: please introduce the abbreviation HFD

Line 83 - 88: I have problems with the formulation: Induction of ERβ / GRP75. I feel that this is too vague. What specifically is induced?? Expression? Activation? Production?

Line 95-97: Why are the homozygous and heterozygous knockout mice named if they are not used?

Line 99: the abbreviation HFD can be introduced earlier (line 74)

Line 112: "(n= 10-15 group) "has to be replaced by "(n = 10-15 animals) ". Furthermore, I am confused why four groups? Shouldn't there be six groups? Female-CTRL, Female-HFD, Female-HFD-CL, Male- CTRL, Male-HFD, and Male-HFD-CL? (additional comment above)

Line 105: this description doesn't fit to figure 1. The pre glucose tolerance test time point is different in M&M and the figure. Please also compare comments on lines 133-140.

Line 125: also, metabolic cages were used at a different age, different to figure 1. The intervention of metabolic cages at the age of 18 weeks is not shown in the figure.

Supplementary fig. 1: "goat anti-rabbit IgG "and "horse anti-mouse IgG " would be correct to name the secondary antibodies!

Line 162: further information concerning the company Biorad can be excluded because it is mentioned already one sentence above.

Supplementary fig. 1: The direction (c- and n-terminal) of the primers is missing. Pgc1a, Leptin, Adiponectin, and Beta-actin are not correct gene names.

Line 202: In my opinion, "H "was not introduced.

Figure 3A: why 17 weeks? Figures H-J are not mentioned in the text and could be transferred to the supplements.

Figure 4a is not mentioned in the text.

Line 233: p = 0.003 or p < 0.003 as mentioned in fig. 4g?

Line 239: basal metabolism? In my opinion, basal metabolism corresponds to the REE, which CL does not influence, but CL increases the TEE!

Figure 5g and 5l are not mentioned in the text.

Line 268: PGC1a (T, p ≤ 0.015, all) … GRP75 (T, p ≤ 0.013, all)

Figure 6g and 7j are not mentioned in the text.

Line 282: For sure, I don´t have the data on the protein expressions in subcutaneous fat but for me, it looks like that the overall treatment effect is higher in females than in males. Did you test the overall treatment effect? Btw the SEM of UCP1 protein expression in SQAT is not within the range of the y-axis. Furthermore, the interruption of the y-axis is unfortunate, because it is not comprehensible which value of PGC1alpha expression is reached in the subcutaneous fat of CL-treated females.

Why are there no significant differences in BAT by CL administration in males? The differences are huge. Is there an overall treatment effect?

Line 284: p = 0.002 or p < 0.002 as mentioned in fig. 9b?

Figure 9g is not mentioned in the text.

In Figures 13 d-i, it is not apparent from the heading or axis designation that AMPK activation or inhibition is being addressed.

Again, the western blot pictures are not mentioned in the text.

Line 336-343: I would suggest using the names of the genes o denote mRNA expression (as used in the figures).

Line 339: The P-value shows no trend of significance in the SQAT of males (fig. 15D; p=0.207)!

Line 361: This figure does not correspond to the M&M description! Please, this part would have to be revised! Measurements at 18 weeks are not included, the duration of HFD is not right, because the last 2 weeks of HFD are combined by CL injections.

Line 363-364: This contradicts the method description! Please, this part would have to be revised!

Line 541-542: citation/literature is missing.

The discussion could be restructured to avoid repeating results (I.e., glucose tolerance, ER) and to acchieve a more clear structure.

Overall, a solid and very comprehensive work. I look forward to the revised version.

Author Response

Dear reviewer 2,

Thank you so much for taking the time to carefully go through our manuscript and for offering such excellent suggestions. I know review work is not compensated and very time consuming, and so we want to express how thankful we are for the time and energy you spent here in helping us to improve the clarity of our manuscript. Our point-by-point responses are as follows:

The authors examined the effect of CL in obesity-induced mice in a sex-specific and tissue-specific manner. The authors examined the animals physiologically (metabolism) on the one hand and the adipose tissues on both protein and RNA levels on the other hand.

The present manuscript contains a large number of results (16. figures, at least from a-e), which on the one hand, is very comprehensive but also has a detrimental effect on the clarity of the manuscript.

Thank you for this comment. We have reduced the number of figures and put some data in tables rather than figures. We do think the manuscript s not much easier to read so really appreciate this reviewers suggestions on condensing and improving the manuscript.

To improve the clarity of the manuscript, I would propose the following:

  1. One of the advantages of publishing with Cells is that you can design the layout of the text and illustrations yourself. Therefore, it would be more than advantageous if the illustrations could be found near the description of the results. Formatting of the paper or a structural adjustment would therefore be desirable.

We had not realized this and thank the reviewer for this excellent suggestion. This improves clarity of the manuscript and certainly reduces the burden on the reader.

  1. Unfortunately, the pictures are not of the best quality. In the meanwhile, I got a proper version, but the lines of the diagrams are sometimes difficult to assign (e.g. Fig. 2a, 2c, 3a, 4a, 5a, etc.). I would prefer a colored representation. I would further suggest enlarging the figures to the whole page width.

Thank you. This was a great suggestion. We re-made all of the figures in color, using the same color scheme throughout. We also enlarged the figures to make them more readable.

  1. For my feeling, the manuscript contains too many illustrations. The included Western blots are not discussed at any time! In my opinion, they could be moved to the supplementary part.

We have moved all of the representative Western blot images to the supplementary section.

I would summarize the PCR data all together in one table because they are rarely significant. However, the figures shown suggest significant differences due to the use of standard errors and the partly unfavorable choice of the Y-axis interruption, which do not exist due to the high deviations between the samples. Nevertheless, I am convinced that there is a significant overall treatment effect, especially for the browning markers. This could all be noted in the table to achieve clarity.

Thank you for this comment. We struggled in determining how to represent those mRNA fold changes. We took the suggestion of moving all PCR data to one table (now Table 1). This had the added benefit of reducing figure number.

  1. Concerning “significances”, I have problems with using letters to mark them here in this manuscript because it suggests also a comparison between i.e. F CL and M CTRL (see Fig. 7a or 7d), which makes no sense. I think it should be more clear to use lines above the bars.

Thank you for pointing out this confusion. Our approach was to perform 2-way ANOVA for sex and treatment and interaction effects. Main effects were indicated with p values on the graphs. In the event when interactions were significant, post-hoc tests were performed to determine between-group differences, which were denoted using the letters. We took the suggestion of getting rid of the letters and simply indicate where interactions occurred, and used bars where appropriate.

  1. At two points, I had difficulty following because the methods section did not introduce me to the respective sections. That was one, the part of gender effect of HFD and CHOW on body weight, etc. Since the methods section only talks about final four groups (line 112) and this comparison is not included in Figure 1, this briefly caused confusion. I would suggest either including the evidence for the effect of HFD in the supplementary part or making additions in the M&M part and/or Figure 1.

The second difficult part was the part about the activation or inhibition of AMPK. There also needs to be an explanation in the methods section.

Thank you for this suggestion. We added language in the methods and results regarding our assessment of AMPK activity.

  1. The Material and methods part should be improved at different places: a) Western Immunoblotting to give more information considering activation and inhibition of AMPK, b) animal treatment to clarify the groups, and c)The method concerning adipocyte cell size is missing.

In addition to describing the AMPK activation assessment, we added language to the methods regarding the groups in order to make the (somewhat complicated) design more clear. We also added methods on cell size analysis – this was an oversight! Thank you.

  1. Furthermore, figure 1 is incomplete and partly contradicts the M&M part (see comment below). This figure would need to be revised.

Done. Thank you!

  1. I found inhomogeneous use of abbreviations for chronic beta3 adrenergic receptor (there are three different kinds of abbreviation) and EchoMRI. Partly too many abbreviations were used anyway. Some like DNL can be saved if the abbreviation was used only once again. The abstract in particular contains 16 abbreviations, some of which were not even introduced. I would suggest to introduce the abbreviations for some of them later.

Excellent suggestions – we have reduced acronym use, made reference to beta3 adrenergic receptor consistent throughout, and edited the abstract as recommended.

  1. Some references in the Introduction are missing.

 Additional references added – thank you for pointing this out.

Furthermore, I have noted mistakes/details that I have recognized (in chronological order) while reading that would also need to be improved (there may be duplications):Thank you so much for this attention to detail – we (and future readers) appreciate your help here.  

Line 2: Replace B3 by Beta-3 like used in the abstract

Done.

Line 19-20: please introduce the abbreviation UCP1, PGC1a, and OxPhos

Done.

Line 62-64: reference is missing

Line 65: literature is missing

Line 67: literature is missing

We added those missing references as suggested – thank you.

Line 74: please introduce the abbreviation HFD

Line 83 - 88: I have problems with the formulation: Induction of ERβ / GRP75. I feel that this is too vague. What specifically is induced?? Expression? Activation? Production?

We have added clarity by indicating whether it was increased protein content or mRNA.

Line 95-97: Why are the homozygous and heterozygous knockout mice named if they are not used?

We changed this to simply “C57BL6 mice”.

Line 99: the abbreviation HFD can be introduced earlier (line 74)

Line 112: "(n= 10-15 group) "has to be replaced by "(n = 10-15 animals) ". Furthermore, I am confused why four groups? Shouldn't there be six groups? Female-CTRL, Female-HFD, Female-HFD-CL, Male- CTRL, Male-HFD, and Male-HFD-CL? (additional comment above)

We hope to have clarified this – thank you.

Line 105: this description doesn't fit to figure 1. The pre glucose tolerance test time point is different in M&M and the figure. Please also compare comments on lines 133-140.

Thank you – we made the appropriate edits for clarity and consistency/accuracy.

Line 125: also, metabolic cages were used at a different age, different to figure 1. The intervention of metabolic cages at the age of 18 weeks is not shown in the figure.

Figure 1 has been corrected.

Supplementary fig. 1: "goat anti-rabbit IgG "and "horse anti-mouse IgG " would be correct to name the secondary antibodies!

Thank you for catching that error – it has been corrected.

Line 162: further information concerning the company Biorad can be excluded because it is mentioned already one sentence above.

Noted and corrected – thank you!

Supplementary fig. 1: The direction (c- and n-terminal) of the primers is missing. Pgc1a, Leptin, Adiponectin, and Beta-actin are not correct gene names.

Thank you for pointing out this error – indeed, our gene names were incorrect and the following corrections were made:

PGC1a gene = Ppargc1a

Leptin gene = Lep

Adiponectin gene = Adipoq

Beta actin gene = Actb

Line 202: In my opinion, "H "was not introduced.

Noted and corrected – thank you!

Figure 3A: why 17 weeks? Figures H-J are not mentioned in the text and could be transferred to the supplements.

We hope our edits to Figure 1 help clarify this. We moved figures to supplementary material and thus reduced the figure number to 11. We are very happy with this change. Thank you!

Figure 4a is not mentioned in the text.

Noted. We opted to remove that figure.

Line 233: p = 0.003 or p < 0.003 as mentioned in fig. 4g?

Noted and corrected – thank you!

Line 239: basal metabolism? In my opinion, basal metabolism corresponds to the REE, which CL does not influence, but CL increases the TEE!

We changed the language here to indicate resting and total energy exp and not ‘basal metabolism’ – we agree that this is more appropriate based on the data. We opted to analyze light cycle EE as an estimate of resting energy expenditure and now report this value along with TEE.  

Figure 5g and 5l are not mentioned in the text.

Noted and corrected. Thanks!

Line 268: PGC1a (T, p ≤ 0.015, all) … GRP75 (T, p ≤ 0.013, all)

Figure 6g and 7j are not mentioned in the text.

Noted. We opted to move those images to supplementary material. Thank you.

Line 282: For sure, I don´t have the data on the protein expressions in subcutaneous fat but for me, it looks like that the overall treatment effect is higher in females than in males. Did you test the overall treatment effect? Btw the SEM of UCP1 protein expression in SQAT is not within the range of the y-axis. Furthermore, the interruption of the y-axis is unfortunate, because it is not comprehensible which value of PGC1alpha expression is reached in the subcutaneous fat of CL-treated females.

We agree that it appears that the directionality is such that the females respond to a greater extent in terms of RNA expression (we assume the reviewer is talking about RNA here?), but this trend did not reach statistical significance. Furthermore, due to the sample sizes for the RNA analyses being smaller, and appreciating that protein expression more accurately represents a functional difference, we put more stock in the protein data. Regarding protein differences, there was no sex by treatment interaction (P=0.444); thus, we do not think it is appropriate to say that the females responded better in SQAT. Yes, we looked at overall treatment effects. In our figures, “T” indicates a main effect of treatment. CL treatment did indeed increase these mitochondrial markers.

As far as the y-axis goes for SQAT UCP1 protein, we think we do need the break, otherwise we cannot see the CTRL levels adequately (that's why we included the break, so we can see all four bars). As this reviewer notes, we did end up cutting off part of the SEM for UCP1. Thus, we reformatted the break in the y-axis so that the entire SEM is in range. Thank you for pointing this out.

Why are there no significant differences in BAT by CL administration in males? The differences are huge. Is there an overall treatment effect?

Yes – indeed there was a significant effect of CL in BAT for both males and females. There was just no main effect of sex, indicated similar expression in males and females.

Line 284: p = 0.002 or p < 0.002 as mentioned in fig. 9b?

Noted and corrected.

Figure 9g is not mentioned in the text.

Noted and corrected.

In Figures 13 d-i, it is not apparent from the heading or axis designation that AMPK activation or inhibition is being addressed.

Noted and corrected.

Again, the western blot pictures are not mentioned in the text.

We moved these to supplement.

Line 336-343: I would suggest using the names of the genes o denote mRNA expression (as used in the figures).

Noted and corrected.

Line 339: The P-value shows no trend of significance in the SQAT of males (fig. 15D; p=0.207)!

Noted and corrected.

Line 361: This figure does not correspond to the M&M description! Please, this part would have to be revised! Measurements at 18 weeks are not included, the duration of HFD is not right, because the last 2 weeks of HFD are combined by CL injections.

Line 363-364: This contradicts the method description! Please, this part would have to be revised!

Noted and corrected. Figure 1 was revised and M&M corrected where necessary.

Line 541-542: citation/literature is missing.

Thank you. We went through and added several references in this place that you identified, and others.

The discussion could be restructured to avoid repeating results (I.e., glucose tolerance, ER) and to acchieve a more clear structure.

We edited the discussion to reduce redundancy – thank you.

Overall, a solid and very comprehensive work. I look forward to the revised version.

Thank you. We very much appreciate your time and constructive suggestions for improvement. The manuscript is much improved as a result.

Reviewer 3 Report

This work compares the response of PGAT, SQAT and BAT to CL injection in female vs. male mice fed a HFD. The topic and the data presented are interesting. However, a few points deserve clarification:

Major comments:

1/ Fig 4F: The use of body-weight adjustment is not recommended when reporting EE data. Please use ANCOVA. (cf. Tschöp et al. Nat Methods. 2011 Dec 28; 9(1): 57–63. & Fernández-Verdejo et al.,Nat Methods 16, 797–799 (2019)). Please consider modifying conclusions afterwards if necessary.

2a/ Figure 8F: I am utterly surprised that the differences between vehicle and CL treatment in male BAT did not reach statistical threshold, with such high differences in gene expressions (PGC1a: over 1000 fold, with low SEM ?!). Can the authors please check the stats again, and comment on that?

2b/ Same goes for cidea (PGAT female, over 10000 fold and not significant?), prdm16, ... (also in other figures, e.g. Fig 15 C,D, list not exhaustive).

3/ If, as one would guess, empirically, a difference in the expression of genes associated with browning is noted in male BAT and not in female BAT, this result should be discussed and part of the conclusion mah have to be amended (line 649-650: " whereas those sex differences do not exist in SQAT or BAT.")

4/ Fig 16: Why isn't BAT depot data presented in this figure? This would be particularly relevant if remark #2a is shown to be true.

5/ Fig 4 B. Energy intake appears to have been miscalculated when expressed as kcal/day (2g/d of a diet worth 4.8 kcal/g energy density would yield an average of ~10 kcal/d, not 1) Fig 4C should be checked too if energy intake (in kcal) was used in the calculation.

6/ Fig4D: Why is TEE presented for 12h light cycle only? there should be a 24-h average report instead, or a 12-h night cycle in parallel.

Minor comments

1/ Introduction, line 74 to 88 (last part): I find it odd to read such part in the introduction, with the main findings summed up. This would fit better in the discussion.

2/ Discussion, line 584: there is probably an unnecessary "that" in the sentence.

Author Response

Dear reviewer 3,

Thank you so much for taking the time to review our work. Your comments were very constructive and appropriate and we hope to have adequately addressed each of them. We do feel that the manuscript is now much improved. We respond below to how we addressed each of your comments:

This work compares the response of PGAT, SQAT and BAT to CL injection in female vs. male mice fed a HFD. The topic and the data presented are interesting. However, a few points deserve clarification:

Major comments:

1/ Fig 4F: The use of body-weight adjustment is not recommended when reporting EE data. Please use ANCOVA. (cf. Tschöp et al. Nat Methods. 2011 Dec 28; 9(1): 57–63. & Fernández-Verdejo et al.,Nat Methods 16, 797–799 (2019)). Please consider modifying conclusions afterwards if necessary.

Noted. This was a great suggestion. We re-analyzed our EE data and report the corrected values. This changed our results such that the sex differences were no longer significant, but there was a main effect of treatment in both sexes. Thank you for this suggestion.

2a/ Figure 8F: I am utterly surprised that the differences between vehicle and CL treatment in male BAT did not reach statistical threshold, with such high differences in gene expressions (PGC1a: over 1000 fold, with low SEM ?!). Can the authors please check the stats again, and comment on that?

Absolutely. We, too, were surprised by the lack of significance and did re-check those. The reason for the lack of significance is the high variability in gene expression. We have noted this in the results section.

2b/ Same goes for cidea (PGAT female, over 10000 fold and not significant?), prdm16, ... (also in other figures, e.g. Fig 15 C,D, list not exhaustive).

Noted – stats were re-checked. See comment above.

3/ If, as one would guess, empirically, a difference in the expression of genes associated with browning is noted in male BAT and not in female BAT, this result should be discussed and part of the conclusion mah have to be amended (line 649-650: " whereas those sex differences do not exist in SQAT or BAT.")4/ Fig 16: Why isn't BAT depot data presented in this figure? This would be particularly relevant if remark #2a is shown to be true.

Unfortunately, the BAT B3AR Western blot was not performed due to time and resource issues. We decided that since we didn't see significant effects  in PGAT and SQAT, that it wasn't necessary to run BAT given all the other data we had. A priori analyses were performed on WAT; the BAT analyses were meant to be supplementary but we ended up producing as much data as possible on this depot in addition to WAT for complete depot comparisons. The reason why we didn't include BAT in the depot comparison is because the aim of the depot comparison was to evaluate the effect of depot on browning, thus we only wanted to compare WAT browning. In other words, we wanted to see what WAT depot was better at browning, NOT whether WAT or BAT was better at browning (that doesn't make sense for what we were addressing). In regards to gene expression, while male values were higher, there was also high variability and lower n’s than in the protein analyses. We extensively looked for outliers but no matter what we did, we couldn't get anything to show up. Thus, we decided to focus more heavily on the protein expression, but felt it appropriate to provide the RT-PCR data for interested readers. In this revision, we opted to report all of the rtPCR data in a table to minimize the number of figures.

5/ Fig 4 B. Energy intake appears to have been miscalculated when expressed as kcal/day (2g/d of a diet worth 4.8 kcal/g energy density would yield an average of ~10 kcal/d, not 1) Fig 4C should be checked too if energy intake (in kcal) was used in the calculation.

Thank you for pointing this out. Indeed, our weekly data had already been averaged to daily kcals, and thus we miscalculated assuming it was weekly and not daily. We have corrected the graphs. We appreciate this reviewer’s careful attention! Thanks again.

 6/ Fig4D: Why is TEE presented for 12h light cycle only? there should be a 24-h average report instead, or a 12-h night cycle in parallel.

Noted and corrected.

Minor comments

1/ Introduction, line 74 to 88 (last part): I find it odd to read such part in the introduction, with the main findings summed up. This would fit better in the discussion.

Noted – a good suggestion. We rearranged this such that the main findings are now summarized in the discussion rather than introduction.

2/ Discussion, line 584: there is probably an unnecessary "that" in the sentence.

Noted – we removed ‘that’ and the sentence now reads better. Thank you!

Round 2

Reviewer 2 Report

The quality and clarity of the manuscript increased greatly after the revision. I still noticed a few things that need to be improved. After that, I would agree with the acceptance of the paper.
1. the dashes above the graphs on the effects of treatments and sex can be removed. This does not look so nice. The explanation below is quite sufficient. I wanted the dashes above the figure with the different fat tissues, because the letters suggest a comparison between, for example, PGAT Control and SQAT CL. But this is nonsensical!
 2. Figure 3 is completely missing. Instead, figure 4 is included twice.
3. In the figure text for Fig 1, read as if it gave the HFD for 16 weeks before CL was administered. Please correct! Line 111
4. I would move the method part "AMPK" to the WIB part.

5. in Fig 4 CTRL is abbreviated differently. There is also some space to enlarge the illustration with the frame.

6. line 232: Figuren 2c!

7. Bring texts together (for example, from page 9 to page 7) to remove unnecessary free space. The text does not necessarily have to be directly above the illustration. It is enough to be close to it.

8. murine genes are not written in italics in my opinion

9. the words table 1a and table 1b are written too small

a few errors are still included: see lines 61, 63,  242, 259, 306, 336, 410, 443, 465, 490, 586

long-term is inconsistently written

4. and one sentence would have to be rewritten. Unfortunately, I do not understand this one. Lines 501-503

Author Response

The quality and clarity of the manuscript increased greatly after the revision. I still noticed a few things that need to be improved. After that, I would agree with the acceptance of the paper.

Thank you for taking the time to carefully review our revised work. Again, your comments were very helpful and extremely constructive.

  1. the dashes above the graphs on the effects of treatments and sex can be removed. This does not look so nice. The explanation below is quite sufficient. I wanted the dashes above the figure with the different fat tissues, because the letters suggest a comparison between, for example, PGAT Control and SQAT CL. But this is nonsensical!

Got it – we misunderstood the comment the first time around. We removed the redundant notation and also like this version much better – thank you.

  1. Figure 3 is completely missing. Instead, figure 4 is included twice.

Noted – thank you! We have fixed this error.

  1. In the figure text for Fig 1, read as if it gave the HFD for 16 weeks before CL was administered. Please correct! Line 111

Noted – we edited the language so that there is no confusion – the mice were fed HFD for 14 weeks prior to treatment, and then 2 weeks during treatment – 16 weeks total. Thank you.

  1. I would move the method part "AMPK" to the WIB part.

Done – We had interpreted this as a request from a previous reviewer but will change this back. Thank you.

  1. in Fig 4 CTRL is abbreviated differently. There is also some space to enlarge the illustration with the frame.

Noted and corrected – Thank you! Looks much better with the larger illustration.

  1. line 232: Figuren 2c!

Noted and corrected – Thank you!

  1. Bring texts together (for example, from page 9 to page 7) to remove unnecessary free space. The text does not necessarily have to be directly above the illustration. It is enough to be close to it.

Noted – this was a journal formatting thing – we submitted figures and legends separately. We will work with the journal on this formatting.

  1. murine genes are not written in italics in my opinion

We changed all genes to non-italics.

  1. the words table 1a and table 1b are written too small

We have edited the tables accordingly. Thank you.

a few errors are still included: see lines 61, 63,  242, 259, 306, 336, 410, 443, 465, 490, 586

61, 63 – 4 new refs added;

242, legend edited;

259, typo found and corrected; minor other edits made;

306, multiple edits made;

336 - 586, many edits made; Thank you for allowing us to more carefully edit our work.

long-term is inconsistently written

We edited the document to make this consistent.

  1. and one sentence would have to be rewritten. Unfortunately, I do not understand this one. Lines 501-503

We decided to delete that sentence. Thank you for the suggestion.

Reviewer 3 Report

My major concern regards the references provided by the authors all throughout the manuscript. Notably (not exhaustive):

Introduction:

  • Line 75 ref 1 and 32 do not tackle any exercise effects on UCP1 (the texts do not even contain the words "exercise" or UCP1)
  • Line 77, likewise, ref 3 and 34 and not related to sex differences in ex-induced fat loss.

Discussion: Ref 61 to 64 are used to justify the effect of CL on AMPK, but none of them are related to this.

I did not check all the references provided but this constitutes a major flaw and is, in my opinion, largely sufficient to recommend rejection of this work for the moment, until this is sorted out.

Other major concern: Fig 4 was pasted twice, instead of Fig 3.

Major concern:

On mitochondria and OXPHOS: Authors should be careful not to extrapolate too far from their data. Example: OXPHOS protein abundance is NOT mitochondrial density.

On AMPK: as stated by authors: "Exercise and fasting are known activators of AMPK in adipose tissue". In this case mice were submitted to a 5h fast without an acute injection of CL before euthanasia. Therefore, here, AMPK activation rather reflects the ability of chronic CL treatment to restore AMPK activation in response to fasting, rather than the effect of CL itself on AMPK. The sentence: "In conclusion, CL promotes WAT AMPK signaling pathways by decreasing the inhibitory phosphoserine..." appears to be inaccurate. 

Minor concerns:

There are a lot of typo mistakes (not exhaustive):

  • line 55 "beneficial"
  • line 63 "treatment"
  • line 575 "In so doing"
  • Line 593 "striking" and "differences"
  • Line 596 "providing"

Figure 9 needs a legend for the colours used

Author Response

My major concern regards the references provided by the authors all throughout the manuscript. Notably (not exhaustive):

Thank you. Many references have been added throughout the manuscript.

Introduction:

  • Line 75 ref 1 and 32 do not tackle any exercise effects on UCP1 (the texts do not even contain the words "exercise" or UCP1)

You are correct – these refs were incorrect and have been corrected – thank you!

  • Line 77, likewise, ref 3 and 34 and not related to sex differences in ex-induced fat loss.

Corrected – thank you.

Discussion: Ref 61 to 64 are used to justify the effect of CL on AMPK, but none of them are related to this.

I did not check all the references provided but this constitutes a major flaw and is, in my opinion, largely sufficient to recommend rejection of this work for the moment, until this is sorted out.

Thank you for pointing out this critical flaw. There may have been an error in adding references using the reference manager. We have thoroughly gone through the entire manuscript, adding missing, or correcting all improperly placed references. The authors thank the reviewers for carefully checking the references. We certainly would not want our work published with incorrect citations.

Other major concern: Fig 4 was pasted twice, instead of Fig 3.

This has been corrected – thank you.

Major concern:

On mitochondria and OXPHOS: Authors should be careful not to extrapolate too far from their data. Example: OXPHOS protein abundance is NOT mitochondrial density.

Thank you for this comment. We implemented this change as to not over-state our findings.

On AMPK: as stated by authors: "Exercise and fasting are known activators of AMPK in adipose tissue". In this case mice were submitted to a 5h fast without an acute injection of CL before euthanasia. Therefore, here, AMPK activation rather reflects the ability of chronic CL treatment to restore AMPK activation in response to fasting, rather than the effect of CL itself on AMPK. The sentence: "In conclusion, CL promotes WAT AMPK signaling pathways by decreasing the inhibitory phosphoserine..." appears to be inaccurate. 

We have changed the sentence to: “A 14-day course of daily CL treatment promoted WAT AMPK signaling pathways by decreasing inhibitory phosphoserine (in the 5-hour fasted state).”

Minor concerns:

There are a lot of typo mistakes (not exhaustive):

  • line 55 "beneficial"
  • line 63 "treatment"
  • line 575 "In so doing"
  • Line 593 "striking" and "differences"
  • Line 596 "providing"

Thank you so much – all have been corrected. For some reason, spell checker had been turned off and we had not realized this! Fixing this allowed us to fix all spelling errors.  

Figure 9 needs a legend for the colours used

Noted and corrected. Thank you.

Round 3

Reviewer 2 Report

Thanks for the resubmission. I accept the current version. I only would suggest reformating fig 1 and fig 2. There are some shifts within the figures.

And one "wight" has remained (fig 1) ;-)

Best wishes!

Author Response

Thank you so much for the time and energy you have invested into reviewing our work, and for your positive comments. We see now that some things happened as we converted PPT figures to PDF images - thank you for pointing this out! We fixed these formatting issues, but cannot find the "wight" in Figure 1?

Reviewer 3 Report

The important flaws of the former version of the manuscript appear to have been corrected, notably regarding the references. I have carefully read this new version and I still came across several issues that require clarification.

  1. Abstract, line 21: As already pointed out "greater mitochondria" is an overstatement since only Oxphos protein abundance was measured.
  2. Figure 1: the figure doesn't read properly in the pdf I have: text is cropped out in the colour rectangles. Also, there are some unwanted line breaks on the bottom that split words in two.
  3. M&M, line 134, authors state that REE was assessed. How was it calculated?
  4. M&M, line 182, an explanation is needed as for the formula used for mRNA expression. The authors appeared to have calculated it using a 2^∆C formula, which is not standard. Can the author develop or add a reference as for this calculation? The formula generally used for determination of gene expression is the method by Kenneth Livak and Thomas Schmittgen in 2001 (known as the 2^–∆∆Ct method), and is quite different from what is stated in the M&M by the authors. For reference:
    https://www.sciencedirect.com/science/article/abs/pii/S1046202301912629?via%3Dihub
    https://toptipbio.com/delta-delta-ct-pcr/
  5. Figure 3: Graph F, appears to show the results of a factorial ANOVA, not ANCOVA as suggested in the sentence line 256 "energy expenditure (T, p=0.004) (Figure 3F), which were adjusted for the effect of body weight (i.e., covaried for effect of body weight (BW))"
    If ANCOVA was used for comparison of 24h-TEE (as it should), the results of the ANCOVA and a graph plotting TEE vs. BW should be presented, at least in suppl fig. The p values of the ANCOVA should appear in the results. The use of ANCOVA should be mentioned in the methods section. Information and free tools for the use of ANCOVA in EE analysis can be found here:
    https://www.mmpc.org/shared/regression.aspx
  6. Figure 5: how was total Oxphos calculated?
  7. Line 311: I could not find oxphos representative blots in suppl. fig.
  8. Discussion: It is a pity that no mention of TEE is made in the discussion as it may be the factor linking together most of the major findings of this study (if ANCOVA confirms the treatment effect). CL would induce browing that would result in increased TEE resulting in turn in lower bw, and possibly improved glucose homeostasis. This should be discussed (if the effects are confirmed in TEE).
  9. Line 522 "adipocyte mitochondrial density" is an overstatement.

Overall, albeit interesting, the manuscript remains hard to read because of the density of data presented. I have 2 suggestions that the author may consider:

A/ The relevance of some of the presented data is debatable, as they seem to add very little to answer the research question  "Compare sex differences in WAT browing markers and ER signalling in relation to mitochondrial function". In order to make the article more concise, the author could question the interest of presenting HSL, ATGL and AMPK data, notably. They are only quickly discussed and no link is make with ERb signalling nor with tissue browning (only a short sentence for AMPK: "has been shown to be required for WAT browning"). The authors should consider either to not show this data (or put it in suppl. fig as an alternative) or develop their relevance in the discussion, with regards to the investigated hypothesis.

B/ A final table summing up the main effects of CL on the different tissues, by sex, could be a good help, with +/-/= symbols to express up/down regulation for example, in the different topics tackled in this work (mitochondrial biogenesis, mitochondrial uncoupling, ERb signalling, etc.)

Author Response

Dear Reviewer, on behalf of all authors, I sincerely thank you for your continued dedication to working with us to best present this work. Your efforts have been visible and much appreciated. The suggestions (especially for the table) were brilliant and you will see that the table was incorporated here. We opted to leave the figures in, but did majorly edit the discussion to hopefully streamline the findings and make the paper more clear and focused. It was an undertaking and there were lots of data to work with, so your help and attention to detail was much appreciated. Thanks again for going above and beyond on this voluntary task of peer review. Best Regards, Vicki

  1. We now say "displayed greater mitochondrial OXPHOS " - thank you.
  2. Thank you - it was a figure conversion thing - It has been corrected.
  3. REE, resting energy expenditure, was calculated using our metabolic chamber system (Sable Systems Promethion) via measurement of VO2 consumed and VCO2 expired in the rodent's inactive (i.e., light) cycle. As suggested, BW was used as a covariate in those analyses of EE. We've added some more detail to the M/M section, which now reads: "

    Indirect calorimetry was utilized before and during CL treatment (i.e., 1 week following the start of treatment). Briefly, animals were placed in indirect calorimetry chambers (Promethion; Sable Systems International, Las Vegas, Nevada) to assess metabolic

    activity parameters including total energy expenditure (TEE), resting energy expenditure (REE), and respiratory quotient (RQ). For TEE and REE, body weight was used as a covariate. Using that same metabolic cage system, spontaneous physical activity (SPA) was measured by the summation of x-, y-, and z-axis beam breaks. Each run captured at least two light and two dark cycles of each variable. Body weight and food intake were recorded on a weekly basis throughout the intervention."

  4. Thank you so much for this attention to detail - you are right - we edited the section accordingly (RNA expression was calculated using the 2-ΔΔCT method, where ΔCT = Housekeeping gene CT - gene of interest CT and presented as fold-difference compared to the reference group.)
  5. Thank you for this comment and resource - We added the following to the results section: "When BW was entered into the ANCOVA model for resting energy expenditure, its effect was significant (p=0.024), as expected. After adjusting for the effect of BW, there was no main effect of sex on resting energy expenditure (p=.793) yet there was a main effect of treatment (p=.001); the SxT interaction was not statistically significant (p=0.109). When BW was entered into the ANCOVA model for total energy expenditure, its effect was again significant (p=0.002), as expected. Again, once adjusting for the effect of BW, there was no main effect of sex (p=0.622) yet there was a main effect of treatment (p=0.004); there was no significant SxT interaction (p=0.144). Males consumed less energy relative to their BW, likely because they were significantly less physically active when compared to females. To summarize, weight loss was induced via CL treatment in both sexes, and this was not explained by reduced energy intake or increased physical activity; rather, it was caused by an increase in non-physical activity energy expenditure." We added the following to the methods: "In order to most accurately assess total and resting energy expenditure (TEE, REE, respectively), we used ANCOVA using body weight as a covariate."
  6. We edited the legend for Figure 5 in the following way: "Perigonadal (PGAT), subcutaneous (SQAT) and brown (BAT) adipose tissue depots were collected and the expression of all five oxidative phosphorylation protein complexes (OXPHOS) were measured via western blot. Complex 1, NADH dehydrogenase (C1). Complex 2, Succinate dehydrogenase (C2). Complex 3, Ubiquinol  Cytochrome c oxidoreductase (C3). Complex 4, Cytochrome c oxidoreductase (C4). Complex 5, ATP Synthase. Total OX PHOS represents the summation of bands C1-5."
  7. Thank you - that figure was indeed accidentally missing from R2; It (ie, WB images) has been added to this revision. 
  8. Thank you for the suggestion. Based on this comment and comment 10, we have edited the discussion.
  9. We substituted the word "proteins" for "density". 
  10. The suggestions for making the manuscript more concise, and adding a table to summarize the sex and depot-specific effects were excellent  and we felt worth addressing. We re-wrote much of the discussion and hope the reviewer agrees that this improves the quality and clarity of the paper. We will certainly respond if the reviewer still think this needs additional work.